# Synchrotron Radiation-Based Refraction-Contrast Tomographic Images Using X-ray Dark-Field Imaging Optics in Human Lung Adenocarcinoma and Histologic Correlations

**DOI:** 10.3390/diagnostics11030487

**Published:** 2021-03-10

**Authors:** Eunjue Yi, Naoki Sunaguchi, Jeong Hyeon Lee, Chul-Yong Kim, Sungho Lee, Sanghoon Jheon, Masami Ando, Yangki Seok

**Affiliations:** 1Department of Thoracic and Cardiovascular Surgery, Korea University Anam Hospital, Seoul 02841, Korea; viking99@hanmail.net (E.Y.); sholeemd@korea.ac.kr (S.L.); 2Department of Radiological and Medical Laboratory Sciences, Graduate School of Medicine, Nagoya University, Nagoya 464-8601, Japan; sunaguchi@met.nagoya-u.ac.jp; 3Department of Pathology, Korea University Anam Hospital, Seoul 02841, Korea; pathjhlee@gmail.com; 4Department of Radiation Oncology, Korea University Anam Hospital, Seoul 02841, Korea; kcyro@korea.ac.kr; 5Department of Thoracic and Cardiovascular Surgery, Seoul National University Bundang Hospital, Seongnam-si 13620, Korea; Jheon@snu.ac.kr; 6Photon Factory, Institute of Materials Structure Science, High Energy Accelerator Research Organization, Tsukuba 300-3256, Japan; ando@post.kek.jp; 7Department of Thoracic and Cardiovascular Surgery, Soonchunhyang University Gumi Hospital, Gumi 39371, Korea

**Keywords:** synchrotron X-ray, lung adenocarcinoma, histology

## Abstract

The aim of this study was to evaluate the clinical implication of synchrotron radiation imaging techniques for human lung adenocarcinoma in comparison with pathologic examination. A refraction-based tomographic imaging technique called the X-ray dark-field imaging (XDFI) method was used to obtain computed tomographic images of human lung adenocarcinoma at the beam line at Photon Factory BL 14B at the High Energy Accelerator Research Organization (KEK) in Tsukuba, Japan. Images of normal lung tissue were also obtained using the same methods and reconstructed as 3D images. Both reconstructed images were compared with pathologic examinations from histologic slides which were made with identical samples. Pulmonary alveolar structure including terminal bronchioles, alveolar sacs, and vasculatures could be identified in synchrotron radiation images of normal lung. Hyperplasia of interstitial tissue and dysplasia of alveolar structures were noticed in images of lung adenocarcinoma. Both synchrotron radiation images were considerably correlated with images from histologic slides. Lepidic patterns of cancer tissue were distinguished from the invasive area in synchrotron radiation images of lung adenocarcinoma. Refraction-contrast tomographic techniques using synchrotron radiation could provide high-resolution images of lung adenocarcinoma which are compatible with those from pathologic examinations.

## 1. Introduction

Lung cancer has been the leading cause of cancer-related death worldwide [1,2]. Patients with early stage disease have showed promising prognosis with 70% to 95% 5-year survival [3]. However, about 75% of lung cancer patients were in advanced stages at the time of diagnosis. Despite the splendid achievement in oncologic management, the survival of those patients still remains in poor; less than 20% of those patients could survive after 1 year from diagnosis [4,5]. Therefore, early detection of lung cancer is essential for survival improvement [1].

Chest computed tomography (CT) is a valuable tool for diagnosis of lung cancer. Adoption of lung cancer screening tests for high-risk patients has been reported to be effective for survival improvement [1]. Although potential additional methods for accelerating the time of early detection have been discussed, such as the detection of biomarkers including proteins or genetic materials derived from early stage diseases using bloods or sputum, currently, chest CT is the rational gold standard with proven survival improvement. With the refinement of chest CT technology, detection of disease could be accelerated at the very early stage [5]. 

Tomographic images of organic tissue using synchrotron radiation have been reported to be superior to conventional imaging modalities [6,7]. This micrometric radiation single beam directed by a monochromator installed in the beamline to receive a white radiation emitted from electrons accelerating at almost the speed of light has been a useful tool, being able to acquire higher-resolution images from small samples of sub-millimeters, even comparable with pathologic features [8]. 

A refraction-based tomographic technique called X-ray dark-field imaging (XDFI) has been evolved to be adapted for deriving internal information from soft tissue, including animal and human specimens, for fifteen years [9]. For biomedical use, a refraction-based tomographic imaging system using synchrotron radiation can show higher-resolution capability with a lower radiation dose compared with conventional tomographic techniques based on absorption contrast theory [7,9,10]. 

We acquired XDFI images from human lung adenocarcinoma specimens at a synchrotron radiation facility called Photon Factory, located in Tsukuba, Japan. The experimental hutch was named BL 14, which was equipped with a vertical wiggler X-ray source and an energy range of 10 to 60 keV (https://www2.kek.jp/imss/pf/eng/apparatus/bl/; accessed on 10 March 2021). By comparing with those from pathologic examinations, we would like to identify the potential of synchrotron radiation imaging techniques for clinical use, including diagnosis.

## 2. Materials and Methods 

### 2.1. Lung Tissue Preparation

A specimen for imaging was obtained from a 67-year-old male patient who had been diagnosed with lung adenocarcinoma (pT1bN0M0, left upper lobe, acinar predominant) and underwent surgical treatment (VATS left upper lobectomy with mediastinal LN dissection) in Korea University Anam Hospital with written informed consent. This study was approved by the Institutional Review Board of Korea University Anam Hospital (IRB number; 2019AN0242).

The resected lung cancer specimen was fixed in 10% neutral buffered formalin solution, and after all pathologic procedures including airway instillation, fixation, gross examination, mapping of cancer, slicing in 3-mm section, paraffin block making, hematoxylin and eosin (H&E) staining, and immunohistochemistry were performed [11], a lung cancer tissue block as well as normal tissue for experiments were harvested under the supervision of a pathologic specialist (J. H. Lee).

The size of the two samples was about 1 × 1.5 cm. Samples for synchrotron radiation imaging were contained in 95% alcoholic solution for transportation and brought to BL 14B of Photon Factory (PF) in the High Energy Accelerator Research Organization (KEK) at Tsukuba, Japan. After image acquisition, those specimens were sent back and prepared as paraffin blocks for pathologic examination.

### 2.2. X-ray Source and Experimental Setup

Synchrotron X-ray was emanated from a 5 Tesla superconducting vertical wiggler installed in the electron storage of PF with an electron acceleration energy of 2.5 GeV and with a typical beam current of 320 mA. 

The optical setting for XDFI imaging was built in BL 14B beamline with a built-in double crystal monochromator with Si (111) diffraction, which produces 19.8 keV. An asymmetrically cut monochromator collimator (AMC) was placed between a sample stage and the double crystal monochromator, which produced a Bragg-reflected X-ray beam. The synchrotron beam passed through splits into two directions by a Laue angle analyzer (LAA), which contained a central area of 172 ± 2–3 μm in thickness and additional cutting of 20 ± 0.2–0.3 μm using chemical etching. This design specification caused a beam from the sample stage to split into two directions: one in the course of the original incident beam (forward direction, dark field) and the other in the direction of diffraction (D, bright field). Charge-coupled device (CCD) cameras in front of the LAA captured the two different types of images, absorption and contrast in a single exposure of X-rays. 

The actual beam size was measured 1160 μm horizontally and 72 μm vertically. The charge-coupled device (CCD) cameras were used to collect imaging signals from the LAA. It has a 14.8 μm pixel size; therefore, the spatial resolution was 15 μm.

### 2.3. Acquisition and Comprehension of Imaging Data

The prepared specimens were transferred from the 95% alcohol solution to a container containing pure water to remove air bubbles from the tissue and replace the alcohol with water. After a sufficient deaeration process, the samples were contained in 1% agarose gel and located in stage for imaging acquisition (Figure 1).

The specimen stage was rotated along the vertical axis precisely during the imaging. The angular interval was 0.3° and the angular span was 180°. During the image acquisition, two sets (forward diffraction and diffraction) of images (600 images each) were obtained. Each projection imaging needed 0.2 seconds for acquisition and the total exposure time was 2 min.

A three-dimensional reconstruction was performed by stacking cross-sectional images reconstructed by the filtered back projection method after convolution of the signum function into each projection, which took approximately 1 hour for each specimen. 

### 2.4. Comparison with Pathologic Examination

After acquisition of tomographic imaging data, those samples were brought back to the hospital and prepared for pathologic examination. Specimens of normal lung parts as well as lung cancer were stained with hematoxylin and eosin (H&E) and observed under a light microscope (LM) by a specialized pathologist. 

## 3. Results

### 3.1. Refraction-Contrast Synchrotron Tomographic Images of Lung Tissue Including Cancer and 3D Reconstruction 

Normal structures included pulmonary vessels, bronchioles, alveolar sacs and interalveolar septa. Intra-alveolar hemorrhage was well identified and distinguished from those findings of lung cancer (Figure 2) in the refraction-contrast images of normal lung tissue. Hemorrhagic lesions inside alveolar spaces are noted as gray to whitish spots in Figure 2b,d. Those lesions could be identified as benign ones because they did not accompany interstitial thickening and preserved their inherent alveolar structures. 

Images from tissue with lung adenocarcinoma showed a well-demarcated cancer area (Figure 3). Three-dimensional tissue images were obtained using a three-dimensional image processing software called the Medical Imaging Interaction Toolkit (MITK), developed by the German Cancer Research Center (Figure 4).

### 3.2. Comparison with Pathologic Examination 

We compared images from LM examinations and those from synchrotron radiation (SR). Intra-alveolar hemorrhages, which were presented as red spots in the H&E image, were observed in the SR image as gray to whitish spots contained in alveolar spaces which were originally filled with air. Images of adenocarcinoma showed thickened interalveolar septa and consolidation, which were clearly distinguished from hemorrhagic lesions in normal lung tissue. The SR images (with a spatial resolution of 15μm) were closely similar with ×100 magnified LM images; however, they could not show distinguishable microstructures as much as a ×400 magnified LM image (Figure 5). 

### 3.3. Special Staining for Diagnosis of Lung Adenocarcinoma Including Immunohistochemistry

For accurate diagnosis and treatment, a special staining method was applied to the lung adenocarcinoma tissue—immunohistochemical (IHC) staining to detect PD-L1 (Programmed death-ligand 1) and ALK (Anaplastic Lymphoma receptor tyrosine Kinase) mutation for our patient. A molecular study was performed to identify ROS (Reactive Oxygen Species) 1 (ROS Proto-Oncogene 1, Receptor Tyrosine Kinase) gene fusion and EGFR (Epidermal Growth Factor Receptor) mutations. The patient showed both PD-L1 and ALK negativity in the immunohistochemical study. ROS1 gene fusion was not found, and EGFR mutation (exon 19 deletion) was found in the molecular study. We did not perform special staining including IHC with the sample we used in imaging acquisition.

## 4. Discussion

In this study, we obtained synchrotron radiation-based refraction-contrast tomographic images using the XDFI method from a lung adenocarcinoma specimen. The resolution of the images was comparable with that of ×100 magnified H&E-stained pathologic images. The X-ray source was a beamline named BL 14B and the optical setting for XDFI was built in the hutch of this beamline. 

Synchrotron radiation X-ray is directly emitted from charged electron particles, with a speed closed to that of light in a linear accelerator [12]. Unlike the ordinary X-ray source requiring tungsten for generation, SR does not need any anode materials. It is characterized by high intensity, tunable energy, polarization and small divergence, can produce superior contrast compared to conventional X-ray methods and, therefore, provides higher spatial resolution up to micrometric scales [6]. Several studies have shown remarkable imaging qualities from human tissue using synchrotron radiation, occasionally comparable with pathologic examination [8,13,14].

However, SR imaging techniques have the inevitable limitation of a small field of view because of its characteristic small divergence [10]. To expand the field of view, beamlines can be elongated extensively to deliver a larger and straightforward beam sufficient for animal or human organs [15]. 

Those long beamline facilities, often called biomedical imaging beamline, have yielded successful tomographic images from human-scale specimens and evolved rapidly, aiming for actual clinical applications [8,15,16]. Clinical trials of microbeam radiation therapy using SR for incurable brain tumor or breast cancer have been in progress with acceptable treatment outcomes [17,18]. Breast cancer screening with synchrotron-based mammography has reported successful clinical results [19].

Those systems usually could be achieved by a sophisticated complex of cutting-edge technology; therefore, they are not easy to construct, even if the entire national capability is concentrated. Currently, only eight long beamline facilities are actively operated worldwide. Each beamline tends to be 100 to 200 m longer than other ordinary ones.

The refraction-based contrast tomographic imaging technique named X-ray dark-field imaging (XDFI) has been well adapted to acquire internal information derived from biomedical soft tissue specimens. By diffraction of X-rays with an asymmetrically cut monochromator collimator (AMC), a larger field of view can be achieved without extensive elongation of the beamline. The acceptance angle in the plan of the AMC improves the X-ray beam divergence from input to output by asymmetry factor, b (Figure 1).

Another peculiar device in this outlay is a special grinded silicon wafer called a Laue angle analyzer (LAA), which has the role of modulating the refractive angle of the X-ray beam refracted by the sample into X-ray intensity, which can produce high contrast in soft tissue. In our study, we could identify the refinement of tomographic images of normal human lung tissue as well as adenocarcinoma using XDFI methods. In our images, the spatial resolution was 15μm because the pixel size of the camera was 14.8μm, and our maximum sample size was 36mm horizontally and 24 mm vertically due to the limitation of the X-ray camera. 

The present status of our beamline capacity cannot be adopted for human-scale imaging; however, it could be applied to small-size samples such as wedge resected lung tissue. If this system could be minimized enough to be installed in a hospital building, it might be used for intraoperative examination of resected specimens without any damage to the specimen. Currently, we have to destroy parts of lung cancer tissue for intraoperative diagnostic examination which we call frozen biopsy, and in cases of very early lung adenocarcinoma, including adenocarcinoma in situ, this damage could interfere the integrity of confirmation. 

However, this study has limitations. Although the resolution of the SR images was comparable with those of ×100 magnified H&E-stained images, it still cannot be applicable for diagnosis because essential characteristics for cancer diagnosis such as cellular hyperproliferation or nuclear duplication could not be observed in the current optical setting, and the results from immunohistochemistry or molecular studies are necessary for diagnosis as well as treatment for lung cancer. Further research to identify specific proteins detected by immunohistochemistry or molecular studies is being conducted using metallic nanoparticles conjugated with specific antibodies [12].

Replication of this experiment could be performed in a restricted environment. It needs synchrotron radiation facilities with a more than 2.5-GeV energy source, a specially designed asymmetrically cut monochromator collimator and a Laue angle analyzer. It also needs computers with specific software which can receive data from CCD cameras, analyze them and reconstruct the data to visual images. The reproduction would be possible only in institutes which can produce synchrotron radiation and equipment for biomedical imaging. This limitation was also one of the technical barriers against the clinical application of SR imaging methods. 

There are still several obstacles for medical adoption; the low accessibility to facilities, the complexity of the optical setting and the narrow field of view. We are expecting to solve parts of those problems with advancements in silicon crystal manufacturing to 450 mm diameter and 1500 mm length. The maximum field of view could be reached at 318 × 318 mm [20]. 

Synchrotron radiation-based XDFI imaging studies have been evaluated in various fields. Several research studies relating breast cancer imaging and calcifications [21,22], imaging of pulmonary diseases including emphysema, fibrosis and lung cancer [23,24] have been reported. One research of dynamic in vivo imaging of mice lung was reported. [25]. With technical refinement of XDFI optics for resolving current limitations, clinical application of synchrotron radiation imaging techniques would be possible. 

## 5. Conclusions

Refraction-based tomographic techniques using synchrotron radiation could provide higher-resolution images of biological soft tissue including lung adenocarcinoma than conventional methods and are often comparable to pathologic examination. With advancements in X-ray optics, clinical application might be achieved in the near future. 

## Figures and Tables

**Figure 1 diagnostics-11-00487-f001:**
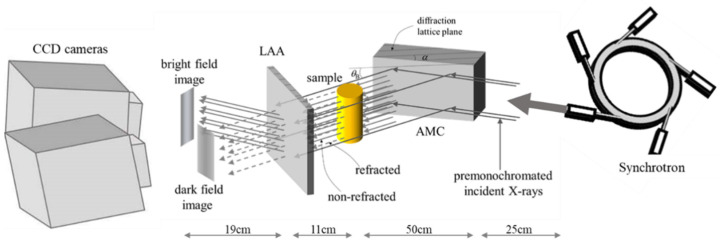
Beamline setup for X-ray dark-field imaging (XDFI) acquisition in BL 14B of Photon Factory. The distance from the beam source to the asymmetrically cut monochromator collimator (AMC) was approximately 25 m, AMC to sample was 50 cm, sample to Laue angle analyzer (LAA) was 11 cm and LAA to camera was 19 cm. Θ_B_ is the Bragg angle, and the angle between the diffracting planes and the crystal surface is α. The acceptance angle of AMC is given by ω_i_ = ω_s_/√b. In this equation, ω_s_ is the intrinsic width of reflection curve in a symmetric case and b is the asymmetric factor calculated according to the following equation: b = sin (Θ_B_ − α)/ sin (Θ_B_ + α) = sin (12.010.2)/sin (12.0 + 10.0) = 0.083, which means the AMC can improve X-ray divergence by a factor b (0.083) [9]. An X-ray beam could suffer from refraction when it passes through the specimen. The refracted beam was forward diffracted by the LAA, which contributed to dark-field imaging.

**Figure 2 diagnostics-11-00487-f002:**
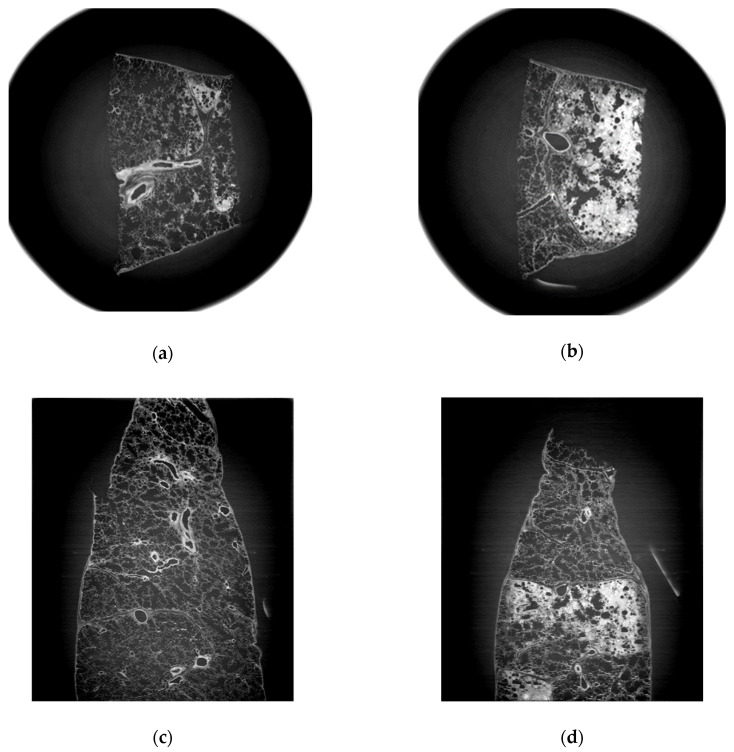
Synchrotron radiation (SR) tomographic images of human normal lung tissue. (**a**) Cross-sectional image of human normal lung tissue. Alveolar spaces, interalveolar septa, bronchioles and microvascular structure are clearly defined. (**b**) Images of normal lung tissue containing hemorrhagic areas inside. They are presented as gray to whitish spots packed in alveolar spaces. Surrounding interstitium was not deformed or thickened. Inherent alveolar structures including microvasculature, interalveolar septa and bronchioles were preserved. (**c**) Sagittal images of normal lung tissue without hemorrhage. (**d**) Sagittal images of normal lung tissue with hemorrhage.

**Figure 3 diagnostics-11-00487-f003:**
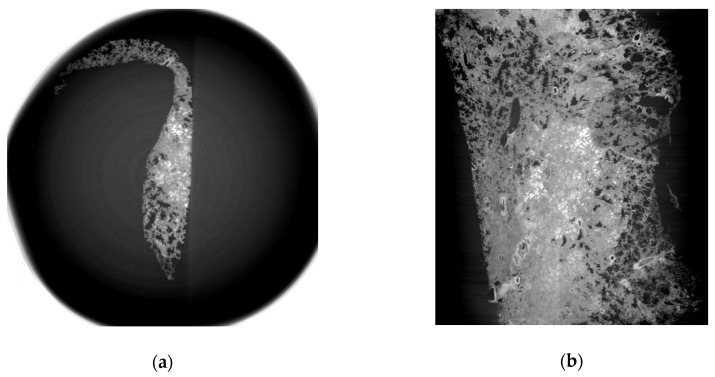
SR tomographic images of lung adenocarcinoma tissue. (**a**) Cross-sectional image of human lung adenocarcinoma tissue. Interstitial thickening and destructed normal alveolar structure including alveolar spaces, interalveolar septa, bronchioles and microvascular structures were observed. (**b**) sagittal images of lung adenocarcinoma. Consolidative lesion with no normal structure was identified. Thickened interstitial tissue and narrowed alveolar spaces were noted.

**Figure 4 diagnostics-11-00487-f004:**
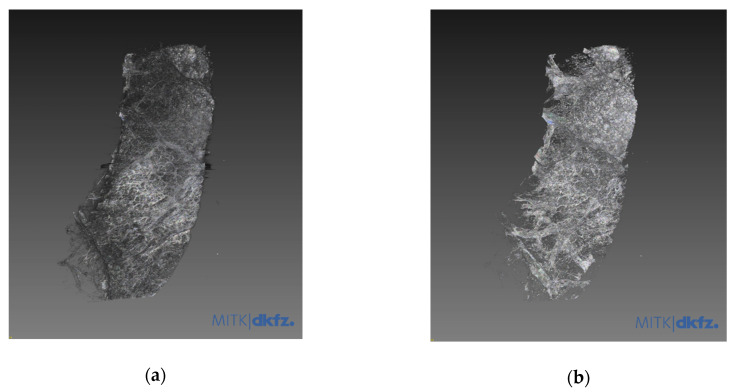
Three-dimensional reconstruction. (**a**) 3D image of normal human lung tissue. (**b**) 3D image of lung adenocarcinoma. Three-dimensional structures of normal lung cancer and adenocarcinoma could be easily observed through 3D reconstruction of tomographic images. The images were obtained using a 3D image processing software called the Medical Imaging Interaction Toolkit (MITK), developed by the German Cancer Research Center. Relating videoclips are appended as Appendix A.

**Figure 5 diagnostics-11-00487-f005:**
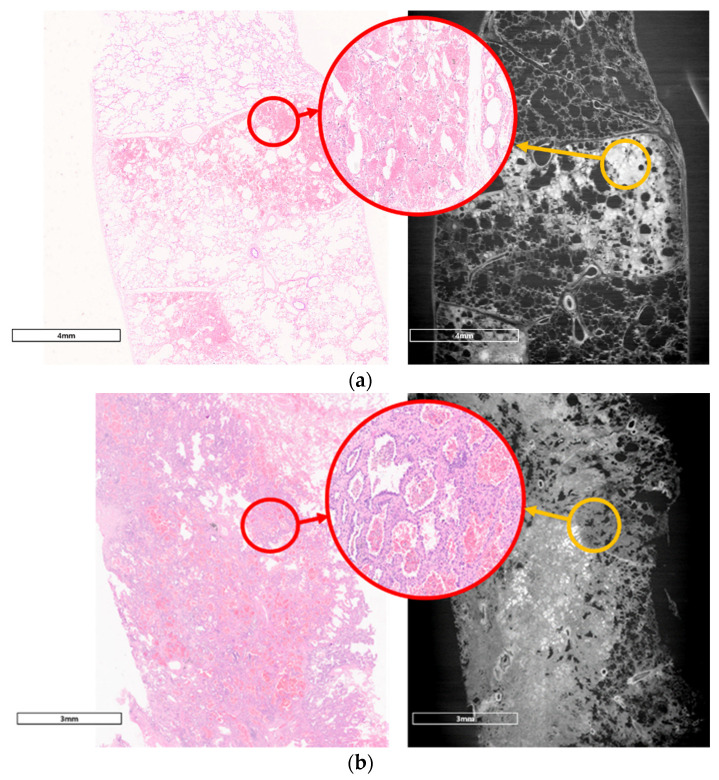
Comparison of pathologic findings and SR images. (**a**) Pathologic image of normal lung tissue and SR image. Hemorrhage inside normal alveolar structure (small red circle) was identified in ×100 magnified image (left side). Normal alveolar structures including alveolar septa and microvasculature were identified in a ×400 magnified image (large red circle in middle). The SR image on the right side is similar to the ×100 magnified light microscope (LM) image. The hemorrhage lesion was identified; however, it was not clear compared with that of the ×400 magnified LM image. (**b**) Pathologic image of lung adenocarcinoma and SR image. Destruction of alveolar structure and interstitial thickening was well noticed both in ×100 magnified LM image (left side) and SR image (right side). However, hyperproliferation and thickening of alveolar lining cells (large red circle in middle) could not be observed in SR image.

## Data Availability

The presented data in this study are openly available.

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
