# Peer review of "Synchrotron Radiation-Based Refraction-Contrast Tomographic Images Using X-ray Dark-Field Imaging Optics in Human Lung Adenocarcinoma and Histologic Correlations"

_diagnostics, 2021, doi:10.3390/diagnostics11030487_

Round 1
Reviewer 1 Report
The manuscript, “Synchrotron Radiation based Refraction-contrast Tomographic images using X-ray Dark-Field Imaging Optics in Human Lung adenocarcinoma and Histologic correlations” by E. Yi, et al. introduced x-ray dark field imaging method to visualize human lung tissue in three-dimension. This technique demonstrates to identify internal structure of the lung tissue. I would like to recommend this manuscript for publication in Diagnostics after revisions. Comments and questions to authors should be addressed;
- Resected tissues were fixed in 10% neutral buffered formalin solution. I wonder if this treatment causes any damages on the tissue.
- The authors described a Laue angle analyzer which provide high contrast in soft tissue. However, there is no evidence if the signal on the detector which collects diffraction signal is used for analysis. The authors should clarify this.
- The authors expect that maximum field of view of x-ray dark filed imaging is 318 mm x 318 mm. Where are these numbers from?
- Description for figures is not sufficient. More detailed information of figures is required.
- I think that the distance between beam source to AMC, sample to LAA, and LAA to camera is not necessary. It is better to mention actual beam size at the sample. I also suggest moving the sentence describing the detector in line 100-102 to section 2.2.
- I suggest swapping figure numbers (Figure 4 and 5) because current Figure 5 is mentioned in this manuscript earlier than the other.
- The scale bar in Figure 4 has to be clear.
- One typo is found, identity in line 208.
Author Response
RESPOND TO THE REVIEWERS
The authors would like to thank Reviewers for careful review of our manuscript and providing us with their comments and suggestion to improve the quality of the manuscript. The following responses have been prepared to address all of the editor’s comments in a point –by-point fashion.
Reviewer 1:
The manuscript, “Synchrotron Radiation based Refraction-contrast Tomographic images using X-ray Dark-Field Imaging Optics in Human Lung adenocarcinoma and Histologic correlations” by E. Yi, et al. introduced x-ray dark field imaging method to visualize human lung tissue in three-dimension. This technique demonstrates to identify internal structure of the lung tissue.
Comment 1:
- Resected tissues were fixed in 10% neutral buffered formalin solution. I wonder if this treatment causes any damages on the tissue.
Answer 1: I and my coauthors sincerely appreciate your comments. Tissue preservation using 10% Formalin solution is general regimen of tissue preparation for pathologic examination. When the resected human specimen is preserved in formalin solution as soon as possible, the progression of damage of tissue could be prevented as much as possible. We can extract genetic materials such as DNA or messenger RNA from tissues in formalin solution, if the preservation process was done fast enough. Therefore, we thought that if there were any damage to the tissue we had used in this study, we though t that it would be because the preservation process was not appropriate, but not because of the formalin solution. We preserved the resected human lung tissue (left upper lobe) in 10% neutral buffered formalin solution as soon as possible, and then cut parts of them (parts of normal lung and parts of lung containing adenocarcinoma, size 1x1.5cm respectively). The specimens we used for imaging acquisition were made into pathologic slide for light microscopic examination, therefore we believe if there were any damage from preservation in formalin solution or preparation process, they would be reflected both in synchrotron radiation image and in pathologic examination.
Changes 2: We revised paragraph related with lung tissue fixation in section 2.1 lung tissue preparation, and added one reference related with lung tissue fixation in the text.
Resected lung cancer specimen was fixed with fixed in 10% neutral buffered formalin solution and after all pathologic procedures including airway instillation, fixation, gross examination, mapping of cancer, slicing with 3mm section, paraffin block making, hematoxylin & eosin (H & E) staining, immunohistochemistry were performed [11], lung cancer tissue block as well as normal tissue for experiment was harvested under supervision of pathologic specialist (J. H. Lee).
[ 11] Hausmann, R. Methods of lung fixation. In Forensic Pathology Reviews, Springer: 2006; pp. 437-451.
Comment 2:
- The authors described a Laue angle analyzer which provide high contrast in soft tissue. However, there is no evidence if the signal on the detector which collects diffraction signal is used for analysis. The authors should clarify this.
Answer 2: We are really grateful for your comments. For increasing contrast, we used Laue angle analyzer which is a specially designed silicon wafer. It looks like a panel of 80mm in diameter, with a central area of 172μm±2-3μm in thickness. We shaved 20μm±0.2-0.3μm from the panel by means of chemical etching at Mimasu semiconductor company (Takasaki, Japan). Those differences of angles in the same plan of analyzer can make the synchrotron beam passes through split into two difference directions. One travels in the dissection of original incident beam, and the other in the direction of diffraction (as we intended with the additional angle of LAA). The CCD cameras catch beams from each direction, and translate the transformation information into one x-ray images. Radiologic signals detected by CCD cameras were collected and transferred to the computer analysis system in beamline hutch and then were made into the actual X-ray graphics. The process of imaging acquisition using synchrotron radiation is quite different from that of conventional x-rays which have been used in medical fields. We used a software program called Medical Imaging Interaction Toolkit (MITK) developed by German Cancer Research Center for acquisition of synchrotron radiation imaging in our study.
We absolutely agreed with the reviewer’s comments that the explanation was not enough. We also thought that the lack of information about our experiment set-up in Figure1. Therefore, we would like to add how the Laue angle analyzer can make high phase contrast images from soft tissue in the text, legend of [Figure 1]. And we would like to change our [Figure 1] for more precise explanation.
Changes 2: We added paragraph in the part of ‘2. Materials and Methods’, section ‘2.2. X-ray source and experimental setup’, we added additional information about LAA and signal analysis. Changes were highlighted with yellowish color in the text.
The synchrotron beam passes through splits into two directions by a Laue Angle Analyzer, which contains a central area of 172μm±2-3μm in thickness and additional cutting of 20μm±0.2-0.3μm using chemical etching. This design specification caused a beam from sample stage split into two directions; one is in the course of original inci-dent beam (forward direction, dark field) and other, direction of diffraction (D, bright field). CCD cameras in front of LAA capture the two different types of images, absorption and contrast in a single exposure of X-rays.
We also changed [Figure 1] as well as its legend.
Figure 1. Beamline setup for XDFI imaging acquisition in BL 14B of Photon Factory. The distance from beam source to AMC was approximately 25m, AMC to sample was 50cm, sample to LAA was 11cm, and LAA to camera was 19cm, respectively. ΘB is Bragg angle and the angle between the diffracting planes and the crystal surface is α. The acceptance angle of AMC is given by ωi=ωs/√b. In this equation, ωs is intrinsic width of reflection curve in a symmetric case and b is asymmetric factor calculated according the equation; b = sin (ΘB − α)/ sin (ΘB + α) = sin (12.010.2)/sin (12.0 + 10.0) = 0.083, which means the AMC can improve X-ray divergence by a factor b (0.083) [9].
[9] Ando, M.; Nakao, Y.; Jin, G.; Sugiyama, H.; Sunaguchi, N.; Sung, Y.; Suzuki, Y.; Sun, Y.; Tanimoto, M.; Kawashima, K. Improving contrast and spatial resolution in crystal analyzer‐based x‐ray dark‐field imaging: Theoretical considerations and experimental demonstration. Medical Physics 2020, 47, 5505-5513.
Comment 3:
- The authors expect that maximum field of view of x-ray dark field imaging is 318 mm x 318 mm. Where are these numbers from?
Answer 3: Thank you for your comments and we deeply apologize for lacking of sufficient explanation. Synchrotron radiation is a monochromatic and micrometric beam which can produce much higher resolution than conventional x-rays do, however, because of the small divergence, human scale imaging has not been produced from synchrotron x-ray source. Numerous studies have been trying to develop novel imaging methods for solving this problem, and our XDFI methods is one of them. Our physicists including professor Ando and Sunaguchi have been devoted to this study for fifteen years.
The world largest diameter of a silicon single crystal is 450 mm so that one can take a largest field of view (FOV) which is equivalent to the inner size of square 450 mm/ √2 = 318mm. This can be a FOV for clinical use when one adopts XDFI to find out early stage of cancer etc. This number has come from the publication from the work of Professor. Ando (Ando, M. et al. Dark-Field Imaging: Recent developments and potential clinical applications. Physica Medica 2016, 32, 1801-1812.) We added this reference in the text. Changes were marked with yellowish color in the text.
Changes 3: We added reference at the end of the sentence that the reviewer had pointed out. Changes were marked with yellowish color in the text.
We are expecting to solve parts of those problems with the advances in silicon crystal manufacturing to 450mm diameter and 1500mm length. The maximum field of view could be reached to 318mm x 318mm [20].
[20] Ando, M.; Sunaguchi, N.; Shimao, D.; Pan, A.; Yuasa, T.; Mori, K.; Suzuki, Y.; Jin, G.; Kim, J.-K.; Lim, J.-H. Dark-Field Imaging: Recent developments and potential clinical applications. Physica Medica 2016, 32, 1801-1812.
Comment 4:
- Description for figures is not sufficient. More detailed information of figures is required.
Answer 4: We are truly grateful for your comments. And absolutely agreed with the reviewer’s opinion. We revised and rewrite the legends of figures to describe sufficient information relating the figures. Changes were highlighted with yellowish color in the text.
Changes 4: We revised the legends of figures according to the reviewer’s comment.
Figure 1. Beamline setup for XDFI imaging acquisition in BL 14B of Photon Factory. The distance from beam source to AMC was approximately 25m, AMC to sample was 50cm, sample to LAA was 11cm, and LAA to camera was 19cm, respectively. ΘB is Bragg angle and the angle be-tween the diffracting planes and the crystal surface is α. The acceptance angle of AMC is given by ωi=ωs/√b. In this equation, ωs is intrinsic width of reflection curve in a symmetric case and b is asymmetric factor calculated according the equation; b = sin (ΘB − α)/ sin (ΘB + α) = sin (12.010.2)/sin (12.0 + 10.0) = 0.083, which means the AMC can improve X-ray divergence by a fac-tor b (0.083) [9].
Figure 2. SR tomographic images of human normal lung tissue. (a) Cross-sectional image of human normal lung tissue. Alveolar spaces, interalveolar septa, bronchioles and microvascular structure are clearly defined. (b) images of normal lung tissue contained hemorrhagic area inside. They were presented as gray to whitish spots packed in alveolar spaces. Surrounding interstitium was not deformed or thickened. Inherent alveolar structures including microvasculature, interalveolar septa and bronchioles were preserved (c) Sagittal images of normal lung tis-sue without hemorrhage. (d) Sagittal images of normal lung tissue with hemorrhage.
Figure 3. SR tomographic images of lung adenocarcinoma tissue. (a) Cross-sectional image of human lung adenocarcinoma tissue. Interstitial thickening and destructed normal alveolar structure including alveolar spaces, interalveolar septa, bronchioles and microvascular structures were observed. (b) sagittal images of lung adenocarcinoma. Consolidative lesion with no normal structure was identified. Thickened interstitial tissue and narrowed alveolar spaces were noted.
Figure 4. Comparison with pathologic findings and SR images. (a) Pathologic images of normal lung tissue and that of SR image. Hemorrhage inside normal alveolar structure (small red circle) was identified in x100 magnified image (left side). Normal alveolar structures including alveolar septa and microvasculature were identified in x400 magnified image (large red circle in middle). The SR image in right side were similar with that of x100 magnified LM image. The hemorrhage lesion was identified however, was not clear compared with that of x400 magnified LM image. (b) Pathologic image of lung adenocarcinoma and that of SR image. Destruction of alveolar structure and interstitial thickening was well noticed both in x100 magnified LM image (left side) and SR image (right side). However, hyperproliferation and thickening of alveolar lining cells (large red circle in middle) could not be observed in SR image.
Figure 5. Three-dimensional reconstruction. (a)3D image of normal human lung tissue. (b) That of lung adenocarcinoma. 3D structures of normal lung cancer and adenocarcinoma could be easily observed through 3D reconstruction of tomographic images. The images were obtained using a 3D image processing software called the Medical Imaging Interaction Toolkit (MITK) developed by German Cancer Research Center. Relating videoclips were appended as supplement data.
Comment 5:
- I think that the distance between beam source to AMC, sample to LAA, and LAA to camera is not necessary. It is better to mention actual beam size at the sample. I also suggest moving the sentence describing the detector in line 100-102 to section 2.2.
Answer 5: We are really grateful for your comments. As you had advised, we removed the sentences about distance from the text. However, we believed that this sentence would be needed for describing details of experimental set-up so we would like to insert those descriptions into the legend of Figure 1.
We added sentences about the actual size of beam we had produced, measured 1160μm horizontally and 72μm vertically. And also we moved the sentence describing detector in line 100-102 to section 2.2 as the reviewer had recommended. Corrections were highlighted with yellowish color in the text.
Changes 5:
We removed the sentences “The distance from beam source to AMC was approximately 25m, AMC to sample was 50cm, sample to LAA was 11cm, and LAA to camera was 19cm, respectively” from the text. And we moved the sentence about detector to section 2.2 as well as adding descriptions of actual beam size according to the reviewer’s advice. Changes were highlighted with yellowish color in the text.
In the section 2.2 X-ray source and experimental setup
The actual beam size was measured 1160μm horizontally and 72μm vertically. The charge-coupled device (CCD) cameras were used to collect imaging signals form LAA. It has 14.8μm pixel size therefore the spatial resolution was 15μm.The charge-coupled device (CCD) cameras were used to collect imaging signals form LAA. It has 14.8μm pixel size therefore the spatial resolution was 15μm.
Comment 6:
- I suggest swapping figure numbers (Figure 4 and 5) because current Figure 5 is mentioned in this manuscript earlier than the other.
Answer 6: I and my coauthors are sincerely grateful for your comments. And we would like to apology for our huge mistake. We should have mentioned Figure 4 before describing Figure 5. Actually, the contents in section ‘2.4. Comparison with pathologic examination’ was a description of Figure 4. We would like to added Figure 4 in this section so that the Figure 4 was mentioned before Figure 5. We are deeply sorry for our errors again.
Changes 6: In the section ‘2.4. Comparison with pathologic examination’, we added Figure 4 in the end of sentence.
After acquisition of tomographic imaging data, those samples were brought back to the hospital and prepared for pathologic examination. Specimen of normal lung parts as well as lung cancer were strained with hematoxylin & eosin (H & E) and observed under a light microscope (LM) by a specialized pathologist (Figure 4).
Comment 7:
- The scale bar in Figure 4 has to be clear
Answer 7: We sincerely appreciate your comments. We revised the pictures in Figure 2, according to the reviewer’s comments.
Changes 7: We revised the scale bar in Figure 4 according to the reviewer’s advice.
Comment 8:
- One typo is found, identity in line 208.
Answer 8: Thank you for your comments. And we deeply apologize for our terrible mistake. We immediately corrected our typo as the reviewer had pointed out.
RESPOND TO THE REVIEWERS
The authors would like to thank Reviewers for careful review of our manuscript and providing us with their comments and suggestion to improve the quality of the manuscript. The following responses have been prepared to address all of the editor’s comments in a point –by-point fashion.
Reviewer 1:
The manuscript, “Synchrotron Radiation based Refraction-contrast Tomographic images using X-ray Dark-Field Imaging Optics in Human Lung adenocarcinoma and Histologic correlations” by E. Yi, et al. introduced x-ray dark field imaging method to visualize human lung tissue in three-dimension. This technique demonstrates to identify internal structure of the lung tissue.
Comment 1:
- Resected tissues were fixed in 10% neutral buffered formalin solution. I wonder if this treatment causes any damages on the tissue.
Answer 1: I and my coauthors sincerely appreciate your comments. Tissue preservation using 10% Formalin solution is general regimen of tissue preparation for pathologic examination. When the resected human specimen is preserved in formalin solution as soon as possible, the progression of damage of tissue could be prevented as much as possible. We can extract genetic materials such as DNA or messenger RNA from tissues in formalin solution, if the preservation process was done fast enough. Therefore, we thought that if there were any damage to the tissue we had used in this study, we though t that it would be because the preservation process was not appropriate, but not because of the formalin solution. We preserved the resected human lung tissue (left upper lobe) in 10% neutral buffered formalin solution as soon as possible, and then cut parts of them (parts of normal lung and parts of lung containing adenocarcinoma, size 1x1.5cm respectively). The specimens we used for imaging acquisition were made into pathologic slide for light microscopic examination, therefore we believe if there were any damage from preservation in formalin solution or preparation process, they would be reflected both in synchrotron radiation image and in pathologic examination.
Changes 2: We revised paragraph related with lung tissue fixation in section 2.1 lung tissue preparation, and added one reference related with lung tissue fixation in the text.
Resected lung cancer specimen was fixed with fixed in 10% neutral buffered formalin solution and after all pathologic procedures including airway instillation, fixation, gross examination, mapping of cancer, slicing with 3mm section, paraffin block making, hematoxylin & eosin (H & E) staining, immunohistochemistry were performed [11], lung cancer tissue block as well as normal tissue for experiment was harvested under supervision of pathologic specialist (J. H. Lee).
[ 11] Hausmann, R. Methods of lung fixation. In Forensic Pathology Reviews, Springer: 2006; pp. 437-451.
Comment 2:
- The authors described a Laue angle analyzer which provide high contrast in soft tissue. However, there is no evidence if the signal on the detector which collects diffraction signal is used for analysis. The authors should clarify this.
Answer 2: We are really grateful for your comments. For increasing contrast, we used Laue angle analyzer which is a specially designed silicon wafer. It looks like a panel of 80mm in diameter, with a central area of 172μm±2-3μm in thickness. We shaved 20μm±0.2-0.3μm from the panel by means of chemical etching at Mimasu semiconductor company (Takasaki, Japan). Those differences of angles in the same plan of analyzer can make the synchrotron beam passes through split into two difference directions. One travels in the dissection of original incident beam, and the other in the direction of diffraction (as we intended with the additional angle of LAA). The CCD cameras catch beams from each direction, and translate the transformation information into one x-ray images. Radiologic signals detected by CCD cameras were collected and transferred to the computer analysis system in beamline hutch and then were made into the actual X-ray graphics. The process of imaging acquisition using synchrotron radiation is quite different from that of conventional x-rays which have been used in medical fields. We used a software program called Medical Imaging Interaction Toolkit (MITK) developed by German Cancer Research Center for acquisition of synchrotron radiation imaging in our study.
We absolutely agreed with the reviewer’s comments that the explanation was not enough. We also thought that the lack of information about our experiment set-up in Figure1. Therefore, we would like to add how the Laue angle analyzer can make high phase contrast images from soft tissue in the text, legend of [Figure 1]. And we would like to change our [Figure 1] for more precise explanation.
Changes 2: We added paragraph in the part of ‘2. Materials and Methods’, section ‘2.2. X-ray source and experimental setup’, we added additional information about LAA and signal analysis. Changes were highlighted with yellowish color in the text.
The synchrotron beam passes through splits into two directions by a Laue Angle Analyzer, which contains a central area of 172μm±2-3μm in thickness and additional cutting of 20μm±0.2-0.3μm using chemical etching. This design specification caused a beam from sample stage split into two directions; one is in the course of original inci-dent beam (forward direction, dark field) and other, direction of diffraction (D, bright field). CCD cameras in front of LAA capture the two different types of images, absorption and contrast in a single exposure of X-rays.
We also changed [Figure 1] as well as its legend.
Figure 1. Beamline setup for XDFI imaging acquisition in BL 14B of Photon Factory. The distance from beam source to AMC was approximately 25m, AMC to sample was 50cm, sample to LAA was 11cm, and LAA to camera was 19cm, respectively. ΘB is Bragg angle and the angle between the diffracting planes and the crystal surface is α. The acceptance angle of AMC is given by ωi=ωs/√b. In this equation, ωs is intrinsic width of reflection curve in a symmetric case and b is asymmetric factor calculated according the equation; b = sin (ΘB − α)/ sin (ΘB + α) = sin (12.010.2)/sin (12.0 + 10.0) = 0.083, which means the AMC can improve X-ray divergence by a factor b (0.083) [9].
[9] Ando, M.; Nakao, Y.; Jin, G.; Sugiyama, H.; Sunaguchi, N.; Sung, Y.; Suzuki, Y.; Sun, Y.; Tanimoto, M.; Kawashima, K. Improving contrast and spatial resolution in crystal analyzer‐based x‐ray dark‐field imaging: Theoretical considerations and experimental demonstration. Medical Physics 2020, 47, 5505-5513.
Comment 3:
- The authors expect that maximum field of view of x-ray dark field imaging is 318 mm x 318 mm. Where are these numbers from?
Answer 3: Thank you for your comments and we deeply apologize for lacking of sufficient explanation. Synchrotron radiation is a monochromatic and micrometric beam which can produce much higher resolution than conventional x-rays do, however, because of the small divergence, human scale imaging has not been produced from synchrotron x-ray source. Numerous studies have been trying to develop novel imaging methods for solving this problem, and our XDFI methods is one of them. Our physicists including professor Ando and Sunaguchi have been devoted to this study for fifteen years.
The world largest diameter of a silicon single crystal is 450 mm so that one can take a largest field of view (FOV) which is equivalent to the inner size of square 450 mm/ √2 = 318mm. This can be a FOV for clinical use when one adopts XDFI to find out early stage of cancer etc. This number has come from the publication from the work of Professor. Ando (Ando, M. et al. Dark-Field Imaging: Recent developments and potential clinical applications. Physica Medica 2016, 32, 1801-1812.) We added this reference in the text. Changes were marked with yellowish color in the text.
Changes 3: We added reference at the end of the sentence that the reviewer had pointed out. Changes were marked with yellowish color in the text.
We are expecting to solve parts of those problems with the advances in silicon crystal manufacturing to 450mm diameter and 1500mm length. The maximum field of view could be reached to 318mm x 318mm [20].
[20] Ando, M.; Sunaguchi, N.; Shimao, D.; Pan, A.; Yuasa, T.; Mori, K.; Suzuki, Y.; Jin, G.; Kim, J.-K.; Lim, J.-H. Dark-Field Imaging: Recent developments and potential clinical applications. Physica Medica 2016, 32, 1801-1812.
Comment 4:
- Description for figures is not sufficient. More detailed information of figures is required.
Answer 4: We are truly grateful for your comments. And absolutely agreed with the reviewer’s opinion. We revised and rewrite the legends of figures to describe sufficient information relating the figures. Changes were highlighted with yellowish color in the text.
Changes 4: We revised the legends of figures according to the reviewer’s comment.
Figure 1. Beamline setup for XDFI imaging acquisition in BL 14B of Photon Factory. The distance from beam source to AMC was approximately 25m, AMC to sample was 50cm, sample to LAA was 11cm, and LAA to camera was 19cm, respectively. ΘB is Bragg angle and the angle be-tween the diffracting planes and the crystal surface is α. The acceptance angle of AMC is given by ωi=ωs/√b. In this equation, ωs is intrinsic width of reflection curve in a symmetric case and b is asymmetric factor calculated according the equation; b = sin (ΘB − α)/ sin (ΘB + α) = sin (12.010.2)/sin (12.0 + 10.0) = 0.083, which means the AMC can improve X-ray divergence by a fac-tor b (0.083) [9].
Figure 2. SR tomographic images of human normal lung tissue. (a) Cross-sectional image of human normal lung tissue. Alveolar spaces, interalveolar septa, bronchioles and microvascular structure are clearly defined. (b) images of normal lung tissue contained hemorrhagic area inside. They were presented as gray to whitish spots packed in alveolar spaces. Surrounding interstitium was not deformed or thickened. Inherent alveolar structures including microvasculature, interalveolar septa and bronchioles were preserved (c) Sagittal images of normal lung tis-sue without hemorrhage. (d) Sagittal images of normal lung tissue with hemorrhage.
Figure 3. SR tomographic images of lung adenocarcinoma tissue. (a) Cross-sectional image of human lung adenocarcinoma tissue. Interstitial thickening and destructed normal alveolar structure including alveolar spaces, interalveolar septa, bronchioles and microvascular structures were observed. (b) sagittal images of lung adenocarcinoma. Consolidative lesion with no normal structure was identified. Thickened interstitial tissue and narrowed alveolar spaces were noted.
Figure 4. Comparison with pathologic findings and SR images. (a) Pathologic images of normal lung tissue and that of SR image. Hemorrhage inside normal alveolar structure (small red circle) was identified in x100 magnified image (left side). Normal alveolar structures including alveolar septa and microvasculature were identified in x400 magnified image (large red circle in middle). The SR image in right side were similar with that of x100 magnified LM image. The hemorrhage lesion was identified however, was not clear compared with that of x400 magnified LM image. (b) Pathologic image of lung adenocarcinoma and that of SR image. Destruction of alveolar structure and interstitial thickening was well noticed both in x100 magnified LM image (left side) and SR image (right side). However, hyperproliferation and thickening of alveolar lining cells (large red circle in middle) could not be observed in SR image.
Figure 5. Three-dimensional reconstruction. (a)3D image of normal human lung tissue. (b) That of lung adenocarcinoma. 3D structures of normal lung cancer and adenocarcinoma could be easily observed through 3D reconstruction of tomographic images. The images were obtained using a 3D image processing software called the Medical Imaging Interaction Toolkit (MITK) developed by German Cancer Research Center. Relating videoclips were appended as supplement data.
Comment 5:
- I think that the distance between beam source to AMC, sample to LAA, and LAA to camera is not necessary. It is better to mention actual beam size at the sample. I also suggest moving the sentence describing the detector in line 100-102 to section 2.2.
Answer 5: We are really grateful for your comments. As you had advised, we removed the sentences about distance from the text. However, we believed that this sentence would be needed for describing details of experimental set-up so we would like to insert those descriptions into the legend of Figure 1.
We added sentences about the actual size of beam we had produced, measured 1160μm horizontally and 72μm vertically. And also we moved the sentence describing detector in line 100-102 to section 2.2 as the reviewer had recommended. Corrections were highlighted with yellowish color in the text.
Changes 5:
We removed the sentences “The distance from beam source to AMC was approximately 25m, AMC to sample was 50cm, sample to LAA was 11cm, and LAA to camera was 19cm, respectively” from the text. And we moved the sentence about detector to section 2.2 as well as adding descriptions of actual beam size according to the reviewer’s advice. Changes were highlighted with yellowish color in the text.
In the section 2.2 X-ray source and experimental setup
The actual beam size was measured 1160μm horizontally and 72μm vertically. The charge-coupled device (CCD) cameras were used to collect imaging signals form LAA. It has 14.8μm pixel size therefore the spatial resolution was 15μm.The charge-coupled device (CCD) cameras were used to collect imaging signals form LAA. It has 14.8μm pixel size therefore the spatial resolution was 15μm.
Comment 6:
- I suggest swapping figure numbers (Figure 4 and 5) because current Figure 5 is mentioned in this manuscript earlier than the other.
Answer 6: I and my coauthors are sincerely grateful for your comments. And we would like to apology for our huge mistake. We should have mentioned Figure 4 before describing Figure 5. Actually, the contents in section ‘2.4. Comparison with pathologic examination’ was a description of Figure 4. We would like to added Figure 4 in this section so that the Figure 4 was mentioned before Figure 5. We are deeply sorry for our errors again.
Changes 6: In the section ‘2.4. Comparison with pathologic examination’, we added Figure 4 in the end of sentence.
After acquisition of tomographic imaging data, those samples were brought back to the hospital and prepared for pathologic examination. Specimen of normal lung parts as well as lung cancer were strained with hematoxylin & eosin (H & E) and observed under a light microscope (LM) by a specialized pathologist (Figure 4).
Comment 7:
- The scale bar in Figure 4 has to be clear
Answer 7: We sincerely appreciate your comments. We revised the pictures in Figure 2, according to the reviewer’s comments.
Changes 7: We revised the scale bar in Figure 4 according to the reviewer’s advice.
Comment 8:
- One typo is found, identity in line 208.
Answer 8: Thank you for your comments. And we deeply apologize for our terrible mistake. We immediately corrected our typo as the reviewer had pointed out.
Changes 8: In line 208, we corrected ‘identity’ to ‘identify’. Corrections were highlighted with yellowish color in the text.
In our study, we can identify the refinement of tomographic images of human normal lung tissue as well as adenocarcinoma using XDFI methods.
Changes 8: In line 208, we corrected ‘identity’ to ‘identify’. Corrections were highlighted with yellowish color in the text.
In our study, we can identify the refinement of tomographic images of human normal lung tissue as well as adenocarcinoma using XDFI methods.
The authors sincerely appreciate the editor’s and reviewers’ valuable comments. We carefully revised our manuscript according to the reviewers’ precious advice. We believe we did our best to improve the quality of our manuscript, and wish our revision have better achievement. If there were anything to be corrected or appended, please let us know and we promise we will make every effort to revise again.

Reviewer 2 Report
This is a proof-of-concept studying using x-ray dark-field imaging (XDFI) techniques on a fully resected lung adenocarcinoma surgical specimen and comparing the XDFI findings with the subsequent pathologic evaluation on light microscopy. This is a very interesting, relatively novel, and important contribution to advance the field of diagnostic imaging however, unfortunately so, there are several limitations that require correction prior to publication. The authors extensively describe modern techniques of biopsy acquisition or post-surgical processing as a significant limitation that interfere with adequate pathologic diagnosis (which I disagree with) but demonstrate a technique that also requires surgical resection of the sample. They did not describe any plans or thoughts on translating their XDFI techniques in vivo to help determine microstructural architecture prior to biopsy or resection, and whether this would even be possible. As a result, using their own logic, this imaging modality does not solve the current problem with our pathologic diagnosis pathway. The XDFI technique and setup details are lacking within the material and methods section and readers would not be able to replicate this study to further explore this field of research based on its current content. As a proof-of-concept study, the potential benefit of having images acquired from the same specimen that is being studied under light microscopy is not fully utilized, either. It is unclear if the specific XDFI findings are being correlated to the same areas seen under microscopy. Additionally, no immunohistochemical (IHC) staining was used for the diagnosis of this patient’s lung adenocarcinoma which does not follow standard of care. The authors should have correlated the microscopic and IHC findings directly with the same areas of the XDFI to better demonstrate its ability to define microstructural anatomy as correlated to the gold standard. Lastly, the discussion is written as an extended introduction with minimal reference to their study result and negligibly contributes to the manuscript impact. Please see below for detailed comments:
- Line 43-45: This statement is false. Lung cancer survival has dramatically increased with advances in treatment. The cited paper saying otherwise is a single author editorial that stands against a vast sea of data. Please refer to the PACIFIC, Checkmate, Keynote, IMpower clinical trials for further information. Additionally, the authors contradict their own statement in the discussion (lines 163-164). This is a very bad way to start a manuscript: providing incorrect statements about lung cancer outcomes and then later contradicting yourself. Please remove this statement entirely.
- Line 74: What surgical procedure was performed? Was this an en bloc resection? What lobe? More details are needed.
- Lines 75-76: What is meant by “were collected carefully after all pathologic procedures for diagnosis were completed”. This needs more detail, both in what “carefully” collecting a specimen means, and what the “pathologic procedures” are.
- Lines 80-83: Does this form of fixation solution impact the architecture of the specimen or the pathologist’s ability to perform routine diagnostic evaluation/IHC staining? If so, it should be mentioned in the limitations. If not, it should be described as a strength in the discussion.
- Line 81: What is BL 14B?
- Line 83: Preparing the specimen for pathologic examination requires more details. Solutions/stains/cuts/IHC/etc.
- Lines 85-93: Need more details on the XDFI setup: tungsten target? Make model of synchrotron or xray source? Anode details? Interferometer used? Micrometer periods? Was a detector used? What make/model detector with details? Details of AMC and LAA. Need details on the optical settings for XDFI. The reader should be able to replicate this study with the details provided in the materials and methods, and this is not currently the case.
- Please label the distances in Figura 1.
- Lines 101-103: image acquisition, processing, reconstruction times?
- Line 103: Typo in “3-dimensional”
- Line 106: typo in title of the section “Comparison”
- Line 110: Was immunohistochemical (IHC) staining used by the pathologist? This is standard of care for lung cancer diagnosis.
- Line 116: Reference to Figure 2 states distinguishing hemorrhagic findings from that of lung cancer, but figure 2 does not show a cancer comparison. Figure 2 shows normal tissue compared to benign hemorrhage, which is an important difference to know. However, the reference to it in Line 116 does not accurately describe it.
- Line 118-120: This should go in the materials/methods.
- Figure 3 is a cross-sectional cancer vs sagittal benign hemorrhage which is not an appropriate comparison. The cancer and benign hemorrhage should be compared on the same image plane, either both cross-sectional or both sagittal to show the difference in imaging techniques and how they correlate.
- Line 124: reference to Figure 4 says intra-alveolar hemorrhage is seen as a “gray shadow in SR image” which is not helpful since the whole image is gray. Figure 4 has a red circle outlining the area of interest in the light microscopy image, it should do the same for the SR image.
- Figure 4: Please also add the 4 and 3 mm ruler legends to the XDFI images so the reader can correlate size comparisons. Are the red circles magnified portions of the light microscopy portion? If so, please describe the magnification amount at baseline, within the red circle, and also add rule legends within the red circle. Please add a red circle on the XDFI image to correlate the same findings of the microscopic image to that of the XDFI finding.
- Results 3.3 should be added to discuss IHC staining for lung adenocarcinoma (as previously mentioned) and correlating those findings with XDFI findings. Otherwise, describe this as a limitation in the discussion.
- The discussion is essentially an extended introduction where the results of the study are minimally discussed. This is poorly written and has no impact or contribution to the manuscript. Please rewrite the discussion section entirely following standard format of: major results of study, why these results are important, other data regarding similar studies, limitations of this study, future outlook, conclusion. The following are comments to the current discussion and should not necessarily be included in the rewritten discussion.
- Lines 154-160: This paragraph contains very strong/bold statements that are false. Invasive carcinoma if often diagnosed on biopsy alone and surgical resection is not necessary. No mention of IHC staining, once again. Please exclude this paragraph entirely from the rewritten discussion.
- Lines 163-164: Contradicts the author’s introductory statements.
- Lines 171-172: This is false. Pathologists do not demand complete resection for a final decision. If AAH, AIS, or MIA are suspected, a pathologist will recommend full surgical resection for full histologic evaluation for areas of invasive carcinoma.
- Lines 175-177: Since the authors mentioned the unnecessary radiation from routine 3–6-month CT Chests, please compare the cGy radiation dose to the patient with routine CT Chests compared to that with XDFI.
- Lines 192-195: Please expand on other XDFI studies. XDFI has been evaluated in breast cancer and calcifications, emphysema, pulmonary fibrosis, there are 2 other lung cancer studies (PMID 28341830 and PMID 24316287) and one on dynamic image capture of the lungs (PMID 30188818)
- The discussion needs a limitations paragraph. Can discuss not having a blinded radiologist to evaluate XDFI images and identify areas of hemorrhage/lung cancer and have the study authors compare those findings to the pathologic areas confirming them. Other limitations are described in my previous comments.
Author Response
RESPOND TO THE REVIEWERS
The authors would like to thank Reviewers for careful review of our manuscript and providing us with their comments and suggestion to improve the quality of the manuscript. The following responses have been prepared to address all of the editor’s comments in a point –by-point fashion.
Reviewer 2:
- The authors described a Laue angle analyzer which provide high contrast in soft tissue. However, there is no evidence if the signal on the detector which collects diffraction signal is used for analysis. The authors should clarify this.
- This is a proof-of-concept studying using x-ray dark-field imaging (XDFI) techniques on a fully resected lung adenocarcinoma surgical specimen and comparing the XDFI findings with the subsequent pathologic evaluation on light microscopy. This is a very interesting, relatively novel, and important contribution to advance the field of diagnostic imaging however, unfortunately so, there are several limitations that require correction prior to publication. The authors extensively describe modern techniques of biopsy acquisition or post-surgical processing as a significant limitation that interfere with adequate pathologic diagnosis (which I disagree with) but demonstrate a technique that also requires surgical resection of the sample. They did not describe any plans or thoughts on translating their XDFI techniques in vivo to help determine microstructural architecture prior to biopsy or resection, and whether this would even be possible. As a result, using their own logic, this imaging modality does not solve the current problem with our pathologic diagnosis pathway. The XDFI technique and setup details are lacking within the material and methods section and readers would not be able to replicate this study to further explore this field of research based on its current content. As a proof-of-concept study, the potential benefit of having images acquired from the same specimen that is being studied under light microscopy is not fully utilized, either. It is unclear if the specific XDFI findings are being correlated to the same areas seen under microscopy. Additionally, no immunohistochemical (IHC) staining was used for the diagnosis of this patient’s lung adenocarcinoma which does not follow standard of care. The authors should have correlated the microscopic and IHC findings directly with the same areas of the XDFI to better demonstrate its ability to define microstructural anatomy as correlated to the gold standard. Lastly, the discussion is written as an extended introduction with minimal reference to their study result and negligibly contributes to the manuscript impact. Please see below for detailed comments:
Comment 1:
- Line 43-45: This statement is false. Lung cancer survival has dramatically increased with advances in treatment. The cited paper saying otherwise is a single author editorial that stands against a vast sea of data. Please refer to the PACIFIC, Checkmate, Keynote, IM power clinical trials for further information. Additionally, the authors contradict their own statement in the discussion (lines163-164). This is a very bad way to start a manuscript: providing incorrect statements about lung cancer outcomes and then later contradicting yourself. Please remove this statement entirely.
Answer 1: I and my coauthors are sincerely grateful for your comments. The survival of lung cancer has been significantly increased as the reviewer had mentioned. We absolutely understand and agreed with the reviewer’s opinion. Our introduction contained too strong expression to provide appropriate introduction. And some of statement contradicts those from the discussion. We changed inadequate sentences and corrected them according to the reviewer’s advice. Changes were highlighted with yellowish color in the text.
Changes 1: We revised our introduction according to the reviewer’s advice. We also changed references that supported our study.
Lung cancer has been leading cause of cancer-related death worldwide [1,2]. Patients with early stage disease have showed promising prognosis with 70 to 95% of 5-year survival [3]. However, about 75% of lung cancer patients were in advanced stages at the time of diagnosis. Despite the splendid achievement in oncologic management, the survival of those patients still remains in poor, less than 20% of those patients could survive after 1 year from diagnosis [4,5]. Therefore, early detection of lung cancer is essential for survival improvement [1].
Chest computed tomography (CT) is a valuable tool for diagnosis of lung cancer. Adoption of lung cancer screening test for high risk patients has been reported to be effective for survival improvement [1]. Although potential additional methods for accelerating the time of early detection have been discussed, such as detection of biomarkers including proteins or genetic materials derived from early stage diseases using bloods or sputum, currently chest CT is rational gold standard with proven survival improvement. With refinement of chest CT technology, detection of disease could be accelerated at the very early stage [5].
[1] Aberle, D.; Adams, A.; Adams, A.; Berg, C.; Black, W.; Clapp, J.; Fagerstrom, R.; Gareen, I.; Gatsonis, C.; Marcus, P. National Lung Screening Trial Research T. Reduced lung-cancer mortality with low-dose computed tomographic screening. N Engl J Med 2011, 365, 395-409.
[2] Lu, T.; Yang, X.; Huang, Y.; Zhao, M.; Li, M.; Ma, K.; Yin, J.; Zhan, C.; Wang, Q. Trends in the incidence, treatment, and survival of patients with lung cancer in the last four decades. Cancer Manag Res 2019, 11, 943-953, doi:10.2147/CMAR.S187317.
[3] Goldstraw, P.; Chansky, K.; Crowley, J.; Rami-Porta, R.; Asamura, H.; Eberhardt, W.E.; Nicholson, A.G.; Groome, P.; Mitchell, A.; Bolejack, V. The IASLC lung cancer staging project: proposals for revision of the TNM stage groupings in the forthcoming (eighth) edition of the TNM classification for lung cancer. Journal of Thoracic Oncology 2016, 11, 39-51.
[4] Walters, S.; Maringe, C.; Coleman, M.P.; Peake, M.D.; Butler, J.; Young, N.; Bergström, S.; Hanna, L.; Jakobsen, E.; Kölbeck, K. Lung cancer survival and stage at diagnosis in Australia, Canada, Denmark, Norway, Sweden and the UK: a population-based study, 2004–2007. Thorax 2013, 68, 551-564.
[5] Blandin Knight, S.; Crosbie, P.A.; Balata, H.; Chudziak, J.; Hussell, T.; Dive, C. Progress and prospects of early detection in lung cancer. Open biology 2017, 7, 170070.
Comment 2:
- Line 74: What surgical procedure was performed? Was this an enbloc resection? What lobe? More details are needed.
Answer 2: We sincerely appreciate your comment, and would like to apology for missing the important information about surgery. The Patient was a 67-year old male with lung adenocarcinoma in his left upper lobe. He was received VATS (Video-assisted thoracoscopic surgery) left upper lobectomy with conventional mediastinal lymph node dissection. We added the surgical details in the text.
Changes 2: We added information about surgical treatment in the text as the reviewer had recommended. Changes were marked with yellowish color in the text.
Specimens for imaging acquisition was obtained from a 67-year old male patient who had been diagnosed as lung adenocarcinoma (pT1bN0M0, left upper lobe, acinar predominant) and underwent surgical treatment (VATS Left upper lobectomy with mediastinal LN dissection) in Korea University Anam Hospital with written informed consent.
Comment 3:
- Lines 75-76: What is meant by “were collected carefully after all pathologic procedures for diagnosis were completed”. This needs more detail, both in what “carefully” collecting a specimen means, and what the “pathologic procedures” are.
Answer 3: Thank you for your comments. We used human lung adenocarcinoma specimen in this study. We believe that the lung cancer tissue resected from patients should not be damaged because of the experimental purpose, so we harvested our specimen after all pathologic procedures for accurate diagnosis were finished. When the surgical resection was performed and lung specimen was extracted from patient’s body, the resected whole specimen was moved to pathologic laboratory which was usually located in next to the operation room in hospitals. Then it was preserved in 10% neutral buffered formalin solution for 3 days. This process is called fixation. Before fixation, the resected lung was inflated with injected air for restoring the actual airway spaces. Fixed lung was carefully examined by a pathologist and this process was reported as ‘gross examination’. A pathologist usually reports his or her description about resected lung specimen, size, location of tumor, and other abnormal findings including invasion of tumor to adjacent structures such as bronchus. Then the lung specimen was sliced serially, generally into 3mm in thickness. The area containing tumor was especially important because the margin status and invasions to the microstructures are important for pathologic reports. H&E stain as well as immunohistochemistry (IHC) were performed. IHCs were usually performed for identifying specific findings such as the presence of EGFR mutation, ALK or k-ras mutations. All those processes were essential for definite diagnosis of lung cancer. Therefore, we would like to harvest our experimental specimen after all these procedures were ended and the pathologist allowed us to get some parts of lung cancer tissue under the patient’s written permission. We would like to express we were doing our best not to make any harmful effect on diagnostic procedures by describing the word ‘carefully’.
However, we deeply agreed that we had to describe our process of specimen harvest. Therefore, we corrected and changed our sentences in section ‘2.1. Lung tissue preparation’. Changes were marked with yellowish color in the text.
Changes 3: We revised the contents describing specimen preparation for experiment according to the reviewer’s advice. Changes were highlighted in yellowish color in the text.
Specimens for imaging acquisition was obtained from a 67-year old male patient who had been diagnosed as lung adenocarcinoma (pT1bN0M0, left upper lobe, acinar predominant) and underwent surgical treatment (VATS Left upper lobectomy with mediastinal LN dissection) in Korea University Anam Hospital with written informed con-sent. This study was approved by Institutional Review Board of Korea University Anam Hospital (IRB number; 2019AN0242).
Resected lung cancer specimen was fixed with fixed in 10% neutral buffered formalin solution and after all pathologic procedures including airway instillation, fixation, gross examination, mapping of cancer, slicing with 3mm section, paraffin block making, hematoxylin & eosin (H & E) staining, immunohistochemistry were performed, lung cancer tissue block as well as normal tissue for experiment was harvested under supervision of pathologic specialist (J. H. Lee).
The size of two samples were about 1x1.5cm respectively. Samples for synchrotron radiation imaging were contained in 95% alcoholic solution for transportation and brought to BL 14B of Photon Factory (PF) in High Energy Accelerator Research Organization (KEK) at Tsukuba, Japan. After imaging acquisition, those specimens were sent back and prepared as paraffin blocks for pathologic examination.
Comment 4:
- Lines 80-83: Does this form of fixation solution impact the architecture of the specimen or the pathologist’s ability to perform routine diagnostic evaluation/IHC staining? If so, it should be mentioned in the limitations. If not, it should be described as a strength in the discussion.
Answer 4: We are grateful for your comments. As we mentioned the above question, the process of fixation using 10% neutral buffered formalin solution was traditional method of pathologists. It might induce tissue damage, however, one of the known best way to preserve tissue. We harvested our samples after all the routine pathologic examinations from fixation to slide review, we believe we did not affect the pathologists’ ability to perform routine diagnostic evaluation. However, when we moved the specimen from formalin to 95% alcohol solution, this process might have influence on the tissue we examined. The specimen might shrink by the high concentrated alcohol solution although it was sufficiently fixed with formalin solution. After finishing imaging acquisition, we brought the specimen back and made them into paraffin block for making pathologic slides. Therefore, we thought that those specimens in imaging and in pathologic examination were identical. However, as the reviewer pointed out, we should mention those limitations in discussion.
Changes 4: We revised and appended the contents relating with pathologic limitation of SR images in the section of discussion. Changes were highlighted with yellowish color in the text.
- Discussion
However, this study has limitation. Although the resolution of SR images was comparable with those of x100 magnified H&E staining images, it still cannot be applicable for diagnosis because essential characteristics for cancer diagnosis such as cellular hyperproliferation or nuclear duplication could not be observed in current optical set-ting, also the results from immunohistochemistry or molecular studies inevitable for diagnosis as well as treatment for lung cancer. Further researches to identify specific proteins detected by immunohistochemistry or molecular studies are being proceeding using metallic nanoparticles conjugated with specific antibodies.
Comment 5:
- Line 81: What is BL 14B?
Answer 5: Thank you for your comments. BL 14B is name of beamline in Photon Factory. Photon Factory is a synchrotron radiation facility located in Tsukuba, Japan, and belongs to High Energy Accelerator Research Organization (KEK) in Tsukuba. This facility contains 22 main beamlines (BL 1-20, 27, and 28). BL 14B is sub beamline of BL B and specialized for high-precision X-ray optics including XDFI imaging methods.
Usually Synchrotron facilities have linear accelerator and storage ring. Accelerated electrons from linear accelerator enter into storage ring and circulate until they are used for experiments. The Photon factory has a 5-Tesla superconducting vertical wiggler installed in the electron storage with an electro acceleration energy of 2.5GeV and with a typical beam current of 320mA. Optical setting for XDFI imaging was built in BL 14B beamline with a built-in double crystal monochromator with Si (111) diffraction and this can attenuate beam energy in to 19.8keV. We would like to add a diagram of Photon factory beam lines.
Changes 5: We added sentences relating with BL 14B in the section of Introduction. Changes were highlighted with yellowish color in the text.
We acquired XDFI images from human lung adenocarcinoma specimen at a synchrotron radiation facility called Photon Factory, located in Tsukuba Japan. The experimental Hutch was named BL 14, which equipped with vertical wiggler X-ray source and energy ranged 10 to 60 keV (https://www2.kek.jp/imss/pf/eng/apparatus/bl/). By comparing with those from pathologic examinations, we would like to identify potential possibility in synchrotron radiation imaging techniques to clinical use including diagnosis.
Comment 6:
- Line 83: Preparing the specimen for pathologic examination requires more details. Solutions/stains/cuts/IHC/etc.
Answer 6: We appreciate your comments. We sincerely agreed with the reviewer’s advice that we should describe more details about pathologic examination. We added descriptions about pathologic examinations with additional reference [11] Hausmann, R. Methods of lung fixation. In Forensic Pathology Reviews, Springer: 2006; pp. 437-451.
Changes 6: In the section ‘2.1 Lung tissue preparation’, we appended additional description of pathologic examination with a reference. Changes were highlighted with yellowish color in the text.
2.1. Lung tissue preparation
Resected lung cancer specimen was fixed with fixed in 10% neutral buffered formalin solution and after all pathologic procedures including airway instillation, fixation, gross examination, mapping of cancer, slicing with 3mm section, paraffin block making, hematoxylin & eosin (H & E) staining, immunohistochemistry were performed, lung cancer tissue block as well as normal tissue for experiment was harvested under supervision of pathologic specialist (J. H. Lee).
Comment 7:
- Lines 85-93: Need more details on the XDFI setup: tungsten target? Make model of synchrotron or x ray source? Anode details? Interferometer used? Micrometer periods? Was a detector used? What make/model detector with details? Details of AMC and LAA. Need details on the optical settings for XDFI. The reader should be able to replicate this study with the details provided in the materials and methods, and this is not currently the case.
Answer 7: Thank you for your comments. The source of synchrotron radiation is electrons accelerated at the speed of light in the linear accelerator of synchrotron radiation facilities. The accelerated electrons enter into storage ring and circulate until they were emitted as light source by bending magnet. As a X-ray source, synchrotron radiation does not need any metals such as tungsten as anode materials. Interferometers are used for synchrotron radiation imaging techniques for sometimes but not always. Much simpler x-ray optics called DEI (Diffraction enhanced imaging) which had been developed in Photon factory used interferometer. Our experiment can be reflected in BL 14B in Photon factory, however, it might not be done in other synchrotron radiation facilities unless they have the same Laue Angle analyzer and Asymmetrically-cut monochromator we have. The energy emitted from light source should be same as we had. It would be sure that our experiment could be not replicated in other ordinary laboratory which have no synchrotron facilities as the reviewer’s comments. We thought that we should describe those limitations in discussion.
Changes 7: We revised our discussion and added limitation relating with replicability of our experiment.
The replication of this experiment could be performed in a restricted environment. It needed synchrotron radiation facilities with more than 2.5GeV energy source, specially designed asymmetrically-cut monochromator collimator and Laue angle analyzer. It also need computers with specific software which can receive data from CCD cameras, analyze and reconstruct to visual images. The reproduction would be possible only in institutes which can produce synchrotron radiation and equipment for biomedical imaging. This limitation was also one of technical barrier against clinical application of SR imaging methods.
Comment 8:
- Please label the distances in Figure 1.
Answer 8: Thank you for your comments. We added labels of the distances we had described in section 2.3. Acquisition and comprehension of imaging data, ‘the distance from beam source to AMC was approximately 25m, AMC to sample was 50cm, sample to LAA was 11cm, and LAA to camera was 19cm’, and we deleted this sentence form the text. We thought that it would be better to avoid duplication of same contents. And we also added explanation about our beamline setup at legend of Figure 1. Changes were marked with yellowish color in the text.
Changes 8: We revised Figure 1 as well as its legend according to the reviewer’s advice. Changes were marked in yellowish color in the text.
Figure 1. Beamline setup for XDFI imaging acquisition in BL 14B of Photon Factory. The dis-tance from beam source to AMC was approximately 25m, AMC to sample was 50cm, sample to LAA was 11cm, and LAA to camera was 19cm, respectively. ΘB is Bragg angle and the angle be-tween the diffracting planes and the crystal surface is α. The acceptance angle of AMC is given by ωi=ωs/√b. In this equation, ωs is intrinsic width of reflection curve in a symmetric case and b is asymmetric factor calculated according the equation; b = sin (ΘB − α)/ sin (ΘB + α) = sin (12.010.2)/sin (12.0 + 10.0) = 0.083, which means the AMC can improve X-ray divergence by a fac-tor b (0.083) [9].
Comment 9:
- Lines 101-103: image acquisition, processing, reconstruction times?
Answer 9: Thank you for your comments. And we do apologize for missing the important information of our experiment. The specimen stage in our experimental setup rotates along the vertical axis. The angular interval is 0.3° and the angular span is 180°. During the imaging acquisition, two sets (forward diffraction and diffraction) of images (600 images each) were obtained. Acquisition of each projection image takes approximately 0.2 to 1.0 second. It takes one second for data transfer from CCD camera to a computer and another one second for saving data in computer. The total data acquisition time for one tomography is 1.5 hours approximately. This acquisition time could be reduced significantly when the imaging data were transferred after completing imaging acquisition using CCD cameras with larger memory capacity.
The process of 3-dimensional reconstruction, which was obtained using 3-dimensional image processing software called the Medical Imaging Interaction Toolkit (MITK) developed by German Cancer Research Center, took approximately 1 hour for each specimen. We added information relating imaging acquisition, processing and reconstruction in the text. Changes were highlighted with yellowish color in the text.
Changes 9: In the section ‘2.3. Acquisition and comprehension of imaging data’, we added sentences describing the time of imaging acquisition, processing and reconstruction according to the reviewer’s advice. Changes were highlighted with yellowish color in the text.
The specimen stage was rotated along the vertical axis precisely during the imaging acquisition. The angular interval is 0.3° and the angular span is 180°. During the imaging acquisition, two sets (forward diffraction and diffraction) of images (600 images each) were obtained. This process took approximately 1.5 hours including data transfer to a computer.
A 3-dimensional reconstruction was performed by stacking cross-sectional images reconstructed by filtered back projection method after the convolution of the signum function into each projection, which took approximately 1 hour for each specimen.
Comment 10:
- Line 103: Typo in “3-dimensional”
Answer 10: We sincerely appreciate your comment, and terribly sorry for our mistakes. We checked the errors we had committed, and corrected them as the reviewer had pointed out. Corrections were highlighted with yellowish color in the text.
Changes 10: We checked the typo in Line 103 ‘A 3-dimentional reconstruction was performed’ and corrected them as ‘A 3-dimensional reconstruction was performed’ according to the reviewer’s comment.
A 3-dimensional reconstruction was performed by stacking cross-sectional images reconstructed by filtered back projection method after the convolution of the signum function into each projection.
Comment 11:
- Line 106: typo in title of the section “Comparison”
Answer 11: I and my coauthors are truly grateful for your comment. And we deeply apologize for our horrible errors. We corrected our mistakes as the reviewer pointed out. Corrections were highlighted with yellowish color in the text.
Changes 11: We checked the typo in Line 103 ‘2.4. Comaparison with pathologic examination’ and corrected them as ‘2.4. Comparison with pathologic examination’
2.4. Comparison with pathologic examination
Comment 12:
- Line 110: Was immunohistochemical (IHC) staining used by the pathologist? This is standard of care for lung cancer diagnosis.
Answer 12: Thank you for your comments. Immunohistochemistry is a widely available technique and essential for lung cancer diagnosis as the reviewer mentioned. Lots of IHC staining methods were used for accurate diagnosis for our specimen including EGFR, ALK and TTF-1. Immunohistochemistry allows for the evaluation of cellular localization of proteins in the context of tumor structure, therefore it is useful for distinguishing lung cancer subtypes or other diseases.
In our experimental setup of XDFI imaging, detection of specific protein localized in tumor tissue was not possible. There had been experiments to detect metal components in breast cancer (Ando 2005), it used fluorescence methods for detecting metallic element. For imaging acquisition comparable with those from IHC staining, there was a necessity for adoption of other synchrotron radiation techniques using nanoparticles. We found one research using nanoparticles for synchrotron radiation imaging and added it as a new reference (Zhu, Y., et al. Synchrotron-based X-ray microscopy for sub-100 nm resolution cell imaging. Current opinion in chemical biology 2017, 39, 11-16).
Changes 12: We revised our discussion and added contents relating with pathologic limitation of SR images in the section of discussion as well as a new reference. Changes were highlighted with yellowish color in the text.
However, this study has limitation. Although the resolution of SR images was comparable with those of x100 magnified H&E staining images, it still cannot be applicable for diagnosis because essential characteristics for cancer diagnosis such as cellular hyperproliferation or nuclear duplication could not be observed in current optical set-ting, also the results from immunohistochemistry or molecular studies inevitable for diagnosis as well as treatment for lung cancer. Further researches to identify specific proteins detected by immunohistochemistry or molecular studies are being proceeding using metallic nanoparticles conjugated with specific antibodies [12].
[12] Zhu, Y.; Zhang, J.; Li, A.; Zhang, Y.; Fan, C. Synchrotron-based X-ray microscopy for sub-100 nm resolution cell imaging. Current opinion in chemical biology 2017, 39, 11-16.
Comment 13:
- Line 116: Reference to Figure 2 states distinguishing hemorrhagic findings from that of lung cancer, but figure 2 does not show a cancer comparison. Figure 2 shows normal tissue compared to benign hemorrhage, which is an important difference to know. However, the reference to it in Line 116 does not accurately describe it.
Answer 13: We sincerely appreciate your comments. We tried to present normal lung tissue and benign lesion, and then lung adenocarcinoma. It would not be appropriate to mention the characteristics of lung cancer in the legend of Figures with normal and benign areas. We corrected our sentences in the text as well as the legend of Figure 2 as the reviewer had pointed out.
Changes 13: We corrected and revised our sentences in the section ‘3.1. Refraction-contrast synchrotron tomographic images of lung tissue including cancer and 3D reconstruction’ as well as in the legend of Figure 2 according to the reviewer’s advice. Changes were marked with yellowish color in the text.
3.1. Refraction-contrast synchrotron tomographic images of lung tissue including cancer and 3D reconstruction
Hemorrhagic lesions inside alveolar spaces were noted as gray to whitish spots in Figure 2-(b) and (d). Those lesions could be identified as benign ones because it did not accompany interstitial thickening and preserved its inherent alveolar structures.
Figure 2. SR tomographic images of human normal lung tissue. (a) Cross-sectional image of human normal lung tissue. Alveolar spaces, interalveolar septa, bronchioles and microvascular structure are clearly defined. (b) images of normal lung tissue contained hemorrhagic area inside. They were presented as gray to whitish spots packed in alveolar spaces. Surrounding interstitium was not deformed or thickened. Inherent alveolar structures including microvasculature, interalveolar septa and bronchioles were preserved (c) Sagittal images of normal lung tissue without hemorrhage. (d) Sagittal images of normal lung tissue with hemorrhage.
Comment 14:
- Figure 3 is a cross-sectional cancer vs sagittal benign hemorrhage which is not an appropriate comparison. The cancer and benign hemorrhage should be compared on the same image plane, either both cross-sectional or both sagittal to show the difference in imaging techniques and how they correlate.
Answer 14: We do appreciate your comment and deeply apologize for our mistakes. The contents of legends figure 2 (b) was not appropriate because this was describing characteristics of benign lesion.
Changes 14: We corrected the legend of Figure 3 according to the reviewer’s advice. Corrections were marked with yellowish color in the text.
Figure 3. SR tomographic images of lung adenocarcinoma tissue. (a) Cross-sectional image of human lung adenocarcinoma tissue. Interstitial thickening and destructed normal alveolar structure including alveolar spaces, interalveolar septa, bronchioles and microvascular structures were observed. (b) sagittal images of lung adenocarcinoma. Consolidative lesion with no normal structure was identified. Thickened interstitial tissue and narrowed alveolar spaces were noted.
Comment 15:
- Line 124: reference to Figure 4 says intra-alveolar hemorrhage is seen as a “gray shadow in SR image” which is not helpful since the whole image is gray. Figure 4 has a red circle outlining the area of interest in the light microscopy image, it should do the same for the SR image.
Answer 15: Thank you for your advice and we completely agreed with the reviewer pointed out. We corrected the description relating with Figure 4 as well as its legend. Changes were marked with yellowish color in the text.
Changes 15: We corrected the sentence in the section 3.2. Comparision with pathologic examination according to the reviewer’s advice. We also corrected the legend of Figure2.
3.2. Comparision with pathologic examination
We compared images from LM examinations and those from synchrotron radiation. Intra-alveolar hemorrhage which were presented as red spots in H & E image, were observed in SR image as gray to whitish spots contained in alveolar spaces which were originally filled with air. Images of adenocarcinoma showed thickened interalveolar septa and consolidation, which were clearly distinguished from hemorrhagic lesions in normal lung tissue (Figure 4).
Figure 2. SR tomographic images of human normal lung tissue. (a) Cross-sectional image of human normal lung tissue. Alveolar spaces, interalveolar septa, bronchioles and microvascular structure are clearly defined. (b) images of normal lung tissue contained hemorrhagic area in-side. They were presented as gray to whitish spots packed in alveolar spaces. Surrounding interstitium was not deformed or thickened. Inherent alveolar structures including microvasculature, interalveolar septa and bronchioles were preserved (c) Sagittal images of normal lung tis-sue without hemorrhage. (d) Sagittal images of normal lung tissue with hemorrhage.
Comment 16:
- Figure 4: Please also add the 4 and 3mm ruler legends to the XDFI images so the reader can correlate size comparisons. Are the red circles magnified portions of the light microscopy portion? If so, please describe the magnification amount at baseline, within the red circle, and also add rule legends within the red circle. Please add a red circle on the XDFI image to correlate the same findings of the microscopic image to that of the XDFI finding.
Answer 16: We really appreciate for your comment. We corrected and revised the pictures in Figure 4. However, we could not add similar SR images with those of red circles in Figure4. We could obtain SR imaging comparable with x100 magnified ones of H&E stain by our SR imaging methods, however, higher resolution comparable with x400 magnification need more research in XDFI imaging method. We deeply apologize that we had not discussed this limitation in the section of discussion. We revised our discussion according to the reviewers’ advice. Corrections were highlighted with yellowish color in the text.
Changes 16: We revised our discussion as well as the legend of Figure 4. according to the reviewer’s advice. Changes were highlighted with yellowish color in the text.
However, this study has limitation. Although the resolution of SR images was comparable with those of x100 magnified H&E staining images, it still cannot be applicable for diagnosis because essential characteristics for cancer diagnosis such as cellular hyperproliferation or nuclear duplication could not be observed in current optical setting, also the results from immunohistochemistry or molecular studies inevitable for diagnosis as well as treatment for lung cancer. Further researches to identify specific proteins detected by immunohistochemistry or molecular studies are being proceeding using metallic nanoparticles conjugated with specific antibodies.
Figure 4. Comparison with pathologic findings and SR images. (a) Pathologic images of normal lung tissue and that of SR image. Hemorrhage inside normal alveolar structure (small red circle) was identified in x100 magnified image (left side). Normal alveolar structures including alveolar septa and microvasculature were identified in x400 magnified image (large red circle in middle). The SR image in right side were similar with that of x100 magnified LM image. The hemorrhage lesion was identified however, was not clear compared with that of x400 magnified LM image. (b) Pathologic image of lung adenocarcinoma and that of SR image. Destruction of alveolar structure and interstitial thickening was well noticed both in x100 magnified LM image (left side) and SR image (right side). However, hyperproliferation and thickening of alveolar lining cells (large red circle in middle) could not be observed in SR image.
Comment 17:
- Results 3.3 should be added to discuss IHC staining for lung adenocarcinoma (as previously mentioned) and correlating those findings with XDFI findings. Otherwise, describe this as a limitation in the discussion.
Answer 17: We are really grateful for your sincere comment. We added resection 3.3 Special staining for diagnosis of lung adenocarcinoma including immunohistochemistry (IHC) according to the reviewer’s comments. We are afraid that we could not obtain SR images similar with IHC, and we are in research to find appropriate methodology for obtaining similar images with those of IHC as we answered previously (In comment 12). We thought we should discuss the limitation relating with IHC in the section of dissection.
Changes 17: We revised and added the section ‘3.3. Special staining for diagnosis of lung adenocarcinoma including immunohistochemistry’. Changes were marked with yellowish color in the text.
3.3. Special staining for diagnosis of lung adenocarcinoma including immunohistochemistry
For accurate diagnosis and treatment, special staining methods was applied to the lung adenocarcinoma tissue. Immunohistochemical (IHC)stain to detect PD-L1 and ALK mutation for our patient. Molecular study was performed to identified ROS1 gene fusion and EGFR mutations. The patient showed both PD-L1 and ALK negative in immunohistochemical study. ROS1 gene fusion was not found, and EGFR mutation (ex-on19 deletion) was found in molecular study). We did not perform special staining including IHC with the sample we used in imaging acquisition.
Comment 18:
- The discussion is essentially an extended introduction where the results of the study are minimally discussed. This is poorly written and has no impact or contribution to the manuscript. Please rewrite the discussion section entirely following standard format of: major results of study, why these results are important, other data regarding similar studies, limitations of this study, future outlook, conclusion. The following are comments to the current discussion and should not necessarily be included in the rewritten discussion.
Answer 18: I and my coauthors are sincerely grateful for your comments and we truly agreed with the reviewer’s advice. We rewrote and reconstruct the whole discussion as the reviewer’s comments.
Changes 18: We revised our discussion according to what the reviewer had pointed out. We rewrote initial part of discussion, removed unnecessary paragraphs. We added content relating with similar studies and study limitations. Corrections were highlighted with yellowish color in the text.
In this study, we obtained Synchrotron radiation based refraction-contrast tomographic images using XDFI method from lung adenocarcinoma specimen. The resolution of images was comparable with that of x100 magnified H&E stained pathologic images. The experiment was performed in a synchrotron radiation facility called Photon factory located in Tsukuba, Japan. The X-ray source was beamline named BL 14B and the optical setting for XDFI imaging was built in the hutch of this beamline.
Synchrotron radiation X-ray is directly emitted from charged electron particles, with a speed closed to that of light in linear accelerator [12]. Unlike the ordinary X-ray source requiring tungsten for generation, SR does not need any anode materials. It is characterized by high intensity, tunable energy, polarization and small divergence, can pro-duce superior contrast to conventional x-ray methods therefore provide higher spatial resolution up to micrometric scales [6]. Several studies have shown remarkable imaging qualities from human tissue using synchrotron radiation occasionally comparable with pathologic examination [8,13,14].
However, SR imaging techniques have inevitable limitation of small field of view because of its characteristic small divergence [10]. To expand the field of view, beam-lines can be elongated extensively to deliver larger and straightforward beam sufficient for animal or human organs [15].
Those long beamline facilities often called biomedical imaging beamline have yielded successful tomographic images from human-scale specimen and evolved rap-idly aiming actual clinical applications [8,15,16]. Clinical trials of microbeam radiation therapy using SR for incurable brain tumor or breast cancer have been in progress with acceptable treatment outcomes [17,18]. Breast cancer screening with synchrotron based mammography had been reported successful clinical results [19].
[19] Castelli, E., et al. A. Mammography with synchrotron radiation: first clinical experience with phase-detection technique. Radiology 2011, 259, 684-694.
Comment 19:
- Lines 154-160: This paragraph contains very strong/boldstatements that are false. Invasive carcinoma if often diagnosed onbiopsy alone and surgical resection is not necessary. No mentionof IHC staining, once again. Please exclude this paragraph entirely from the rewritten discussion.
Answer 19: We sincerely appreciate your comments. We also absolutely agreed with the reviewer’s advice. We thought that the first paragraph was not necessary. We removed all contents that the reviewer had pointed out and added the limitation relating with immunohistochemistry.
Changes 19: We removed the first paragraph and added limitation relating with immunohistochemistry study in the discussion. The first paragraph was eased, and the limitation sentences were highlighted with yellowish color in the text.
It is essential for accurate diagnosis of early stage of lung adenocarcinoma to preserve whole specimen with cancer area without any destruction including biopsy [10]. However, in reality, there is no way to confirm diagnosis without damages unless undergoing surgical procedure [16]. Neither percutaneous needle biopsy nor frozen procedure can be performed preserving the whole cancer tissue intact. Currently we have to impair the integrity of suspicious lesion for providing pre-surgical information to healthcare relatives, patients as well as clinicians.
However, this study has limitation. Although the resolution of SR images was comparable with those of x100 magnified H&E staining images, it still cannot be applicable for diagnosis because essential characteristics for cancer diagnosis such as cellular hyperproliferation or nuclear duplication could not be observed in current optical setting, also the results from immunohistochemistry or molecular studies inevitable for diagnosis as well as treatment for lung cancer. Further researches to identify specific proteins detected by immunohistochemistry or molecular studies are being proceeding using metallic nanoparticles conjugated with specific antibodies [12].
Comment 20:
- Lines 163-164: Contradicts the author’s introductory statements
Answer 20: We really appreciate your comments. We totally agreed with the reviewer’s opinion that this statement was contradict to our introduction. We removed the sentence and rewrote the introduction as well as the first part of discussion.
Changes 20: We removed the unnecessary statement that the reviewer had pointed out and rewrote the introduction. Changes were highlighted with yellowish color in the text.
It obviously contributes to improve lung cancer survival, nevertheless has been provoked novel challenges of overdiagnosis or excessive treatment
Introduction
Lung cancer has been leading cause of cancer-related death worldwide [1,2]. Patients with early stage disease have showed promising prognosis with 70 to 95% of 5-year survival [3]. However, about 75% of lung cancer patients were in advanced stages at the time of diagnosis. Despite the splendid achievement in oncologic management, the survival of those patients still remains in poor, less than 20% of those patients could survive after 1 year from diagnosis [4,5]. Therefore, early detection of lung cancer is essential for survival improvement [1].
Chest computed tomography (CT) is a valuable tool for diagnosis of lung cancer. Adoption of lung cancer screening test for high risk patients has been reported to be effective for survival improvement [1]. Although potential additional methods for accelerating the time of early detection have been discussed, such as detection of biomarkers including proteins or genetic materials derived from early stage diseases using bloods or sputum, currently chest CT is rational gold standard with proven survival improvement. With refinement of chest CT technology, detection of disease could be accelerated at the very early stage [5].
Comment 21:
- Lines 171-172: This is false. Pathologists do not demand complete resection for a final decision. If AAH, AIS, or MIA are suspected, a pathologist will recommend full surgical resection for full histologic evaluation for areas of invasive carcinoma
Answer 21: Thank you for your precious comment and we do agree with the reviewer’s advice. The paragraph relating with surgical resection for diagnosis seemed to be inappropriate and unnecessary. We would like to remove all the sentences as the reviewer had recommended.
Changes 21: We would like to remove the paragraph that the reviewer had pointed out. Because whole part of discussion was reconstructed, those sentences were disappeared in the revised manuscript.
Diagnostic uncertainty in early stage lung cancer could induce overuse of invasive procedures. Precancerous lesion known as atypical adenomatous hyperplasia (AAH) and early cancer state including adenocarcinoma in situ (AIS) or minimally invasive adeno-carcinoma (MIA) are frequently indistinguishable because all those lesions can be appeared as ground glass opacity in chest CT. Pathologists demand complete histological sampling for final decision. It is more important when the tumor is adenocarcinoma in situ, because they need to identify noninvasiveness entirely. Paradoxically, the smaller the nodule is, surgical intervention was inevitable for assuring tumor status.
Comment 22:
- Lines 175-177: Since the authors mentioned the unnecessary radiation from routine 3–6-month CT Chests, please compare the cGy radiation dose to the patient with routine CT Chests compared to that with XDFI
Answer 22: Thank you for your comments. The radiation we had used in our experiment was 19.8keV, which contained very low energy because the 1eV=1.6x10-19J. Therefore, the radiation level in our experiment could be safe for living animal. However, the field of view that the X-ray was irradiated so we could not sure the safety for human beings. The photon energy of conventional x-ray method in clinical field are higher than 19.8keV. Safety check system for the secure of patients should be considered if synchrotron radiation imaging methods were adopted in clinical application.
|
Use |
Accelerating potential |
Target |
Average |
|
|
Dianostic |
Mammography |
26 - 30 kV |
Rhodium Molybdenum |
20 keV |
|
Dental |
60 kV |
Tungsten |
30 keV |
|
|
General |
50 - 140 kV |
Tungsten |
40 keV |
|
|
CT |
80 - 140 kV |
Tungsten |
60 keV |
|
|
Baggage screening |
80 – 160 kV |
Tungsten |
80 keV |
|
Changes 21: We revised our discussion and removed the initial several paragraphs. Therefore, the sentence that the reviewer had pointed out has been removed. We do apologize that we could not carry out what the reviewer had mentioned.
Comment 23:
- Lines 192-195: Please expand on other XDFI studies. XDFI has been evaluated in breast cancer and calcifications, emphysema, pulmonary fibrosis, there are 2 other lung cancer studies (PMID28341830 and PMID 24316287) and one on dynamic image capture of the lungs (PMID 30188818)
Answer 23: Thank you for your recommendation. We revised our discussion and added what the reviewer had suggested in the text.
Changes 23: We revised the sentences and added references that the reviewer had suggested. Changes were marked with yellowish color in the text.
Synchrotron radiation based XDFI imaging studies have been evaluated in varies fields. Several researches relating breast cancer imaging and calcifications [21,22], imaging of pulmonary diseases including emphysema, fibrosis, and lung cancer [23,24] have been reported. One research of dynamic in vivo imaging of mice lung was reported. [25].
[21] Ando, M.; Yamasaki, K.; Ohbayashi, C.; Esumi, H.; Hyodo, K.; Sugiyama, H.; Li, G.; Maksimenko, A.; Kawai, T. Attempt at two-dimensional mapping of X-ray fluorescence from breast cancer tissue. Japanese journal of applied physics 2005, 44, L998.
[22] Ando, M.; Sunaguchi, N.; Wu, Y.; Do, S.; Sung, Y.; Louissaint, A.; Yuasa, T.; Ichihara, S.; Gupta, R. Crystal analyser-based X-ray phase contrast imaging in the dark field: implementation and evaluation using excised tissue specimens. European radiology 2014, 24, 423-433.
[23] Willer, K.; Fingerle, A.A.; Gromann, L.B.; De Marco, F.; Herzen, J.; Achterhold, K.; Gleich, B.; Muenzel, D.; Scherer, K.; Renz, M. X-ray dark-field imaging of the human lung—A feasibility study on a deceased body. PLoS One 2018, 13, e0204565.
[24] Meinel, F.G.; Schwab, F.; Yaroshenko, A.; Velroyen, A.; Bech, M.; Hellbach, K.; Fuchs, J.; Stiewe, T.; Yildirim, A.Ö.; Bamberg, F. Lung tumors on multimodal radiographs derived from grating-based X-ray imaging–A feasibility study. Physica Medica 2014, 30, 352-357.
[25] Gradl, R.; Morgan, K.; Dierolf, M.; Jud, C.; Hehn, L.; Günther, B.; Möller, W.; Kutschke, D.; Yang, L.; Stoeger, T. Dynamic in vivo chest x-ray dark-field imaging in mice. IEEE transactions on medical imaging 2018, 38, 649-656.
Comment 24:
- The discussion needs a limitations paragraph. Can discuss not having a blinded radiologist to evaluate XDFI images and identify areas of hemorrhage/lung cancer and have the study authors compare those findings to the pathologic areas confirming them. Other limitations are described in my previous comments.
Answer 24: We sincerely appreciate your comments. We do agree with the reviewer’s advice. Since this study was from experimental setup in a synchrotron radiation facility, it had lots of limitation especially for clinical application. We described limitations relating with our experiment in the end of discussion as the reviewer had suggested.
Changes 24: We revised our discussion and inserted sentences relating with our study limitation in the discussion according to the reviewer’ advice. Corrections were marked with yellowish color in the text.
However, this study has limitation. Although the resolution of SR images was comparable with those of x100 magnified H&E staining images, it still cannot be applicable for diagnosis because essential characteristics for cancer diagnosis such as cellular hyperproliferation or nuclear duplication could not be observed in current optical setting, also the results from immunohistochemistry or molecular studies inevitable for diagnosis as well as treatment for lung cancer. Further researches to identify specific proteins detected by immunohistochemistry or molecular studies are being proceeding using metallic nanoparticles conjugated with specific antibodies [12].
The replication of this experiment could be performed in a restricted environment. It needed synchrotron radiation facilities with more than 2.5GeV energy source, specially designed asymmetrically-cut monochromator collimator and Laue angle analyzer. It also needs computers with specific software which can receive data from CCD cameras, analyze and reconstruct to visual images. The reproduction would be possible only in institutes which can produce synchrotron radiation and equipment for biomedical imaging. This limitation was also one of technical barrier against clinical application of SR imaging methods.
There are still several obstacles for medical adoption; the low accessibility to the facilities, complexity of optical setting and narrow field of view. We are expecting to solve parts of those problems with the advances in silicon crystal manufacturing to 450mm diameter and 1500mm length. The maximum field of view could be reached to 318mm x 318mm [20].
The authors sincerely appreciate the editor’s and reviewers’ valuable comments. We carefully revised our manuscript according to the reviewers’ precious advice. We believe we did our best to improve the quality of our manuscript, and wish our revision have better achievement. If there were anything to be corrected or appended, please let us know and we promise we will make every effort to revise again.

Round 2
Reviewer 1 Report
I appreciate all of author’s efforts to improve this manuscript. This revised manuscript becomes clearer than the original. I would like to recommend this manuscript for publication in Diagnostics after revisions. Comments to authors should be addressed;
- It looks better to mention actual exposure time at each scan points in section 2.3.
- It is better to state regarding resolution of SR imaging (x100) compared to pathologic images in section 3.2. It supports this work, but it appears only in the caption of Figure 4.
- In Figure 1, authors indicated refracted and non-refracted. I can’t get what these meant.
- The order of figures in the current version is messed up so that readers need to turn pages back and forth to find figures. The current order does not look smooth and I think it is better to show all images regarding SR imaging first and then the one on comparison. The following is my suggestion;
Figure 1. Beamline setup for XDFI imaging acquisition in BL 14B of Photon Factory
Figure 2. SR tomographic images of human normal lung tissue
Figure 3. SR tomographic images of lung adenocarcinoma tissue
Figure 4. Three-dimensional reconstruction.
Figure 5. Comparison with pathologic findings and SR images
Authors need to check figure numbers in the manuscript and some of them can be removed, e.g. (Figure 4) in line 130.
- Line 205 to 207 is not necessary to mention in discussion because it is mentioned in section 2.
- Line 134-135 doesn’t look like a complete sentence.
- One typo is found, form in line 111.
Author Response
RESPOND TO THE REVIEWERS
The authors would like to sincerely appreciate the Reviewers for careful review of our manuscript and providing us with their comments and suggestion. Those valuable comments were tremendously helpful for improving the quality of our manuscript. We carefully revised our previous manuscript to adjust it according to the reviewers’ advices. The following responses have been prepared to address all of the editor’s comments in a point –by-point fashion.
Reviewer 1:
I appreciate all of author’s efforts to improve this manuscript. This revised manuscript becomes clearer than the original. I would like to recommend this manuscript for publication in Diagnostics after revisions. Comments to authors should be addressed;
Comment 1:
- It looks better to mention actual exposure time at each scan points in section 2.3.
Answer 1: I and my coauthors are sincerely grateful for your comments. It took 0.2 second for acquisition of each projection and total number of projection imaging were 600, therefore exposure time were 2 minutes. We added the sentence describing exposure time in the text as the reviewer had mentioned.
Changes 1: We inserted the sentence describing total exposure time in the text according to the reviewer’s advice. Changes were highlighted with yellowish color in the text.
Each projection imaging needed 0.2second for acquisition and total exposure time were 2 minutes. This process took approximately 1.5 hours including data transfer to a computer.
Comment 2:
- It is better to state regarding resolution of SR imaging (x100) compared to pathologic images in section 3.2. It supports this work, but it appears only in the caption of Figure 4.
Answer 2: We are extremely thankful for your comments. We should have inserted sentences relating the resolution of SR and LM images. We added sentence stating resolution of SR as well as pathologic images as the reviewer had suggested.
Changes 2: We inserted sentence describing the resolution of SR and pathologic images in the section 3.2. Comparison with pathologic examination according to the reviewer’s advice. Changes were marked with yellowish color in the text.
The SR images (with spatial resolution of 15μm) were closely similar with x100 magni-fied LM images, however, could not show distinguishable microstructures as much as x400 magnified LM image (Figure 5).
Comment 3:
- In Figure 1, authors indicated refracted and non-refracted. I can’t get what these meant.
Answer 3: Thank you for your comments, and we do apologize that we had not describe the legend of Figure 1 properly. The asymmetrically-cut monochromator collimator (AMC) could afford to split the x-ray beam according to the angle of incidence, thereby increasing the size of beam. Enlarged beam could be refracted when it passes through the specimen according to the density of sample, which is one of characteristics of light. Laue angle Analyzer (LAA) can pick up refracted x-ray beam under high contrast and resolution, which was called forwardly diffracted beam. In the absence of specimen, most beam was not refracted and the diffracted by LAA could have maximum intensity, therefore the beam was called bright field. We would like to insert sentences relating with refracted beam and dark field in the legend of Figure 1 as the reviewer had mentioned.
Changes 3: We added the sentence relating with refraction of beam in the legend of Figure 1 according to the reviewer’s comments. Changes were highlighted with yellowish color in the text.
X-ray beam could suffer from refraction when passes through the specimen. Refracted beam was forward diffracted by LAA, which was contributed to dark field imaging.
Comment 4:
- The order of figures in the current version is messed up so that readers need to turn pages back and forth to find figures. The current order does not look smooth and I think it is better to show all images regarding SR imaging first and then the one on comparison. The following is my suggestion;
Figure 1. Beamline setup for XDFI imaging acquisition in BL 14B of Photon Factory
Figure 2. SR tomographic images of human normal lung tissue
Figure 3. SR tomographic images of lung adenocarcinoma tissue
Figure 4. Three-dimensional reconstruction.
Figure 5. Comparison with pathologic findings and SR images
Authors need to check figure numbers in the manuscript and some of them can be removed, e.g. (Figure 4) in line 130.
Answer 4: We truly appreciate for your comments. The figures would be more concordant with the contents describing figures in the text if we changed the orders according to the reviewer had suggested. We thought that we could describe the results of our experiment more clearly and rationally after we corrected the order of figures as the reviewer had mentions. We also realized that we did not need to add the ‘figure 4’ at Section 2.3 Acquisition and comprehension of imaging data to let Figure 4 come out prior to Figure 5.
Changes 4: We changed the order of figures according to the reviewer had suggested. We also remove the phrase ‘Figure 4’ from line 130, and reflected the changed number of figures in the text. Changes were marked with yellowish color in the text.
Specimen of normal lung parts as well as lung cancer were strained with hematoxylin & eosin (H & E) and observed under a light microscope (LM) by a specialized pathologist (Figure 4).
Images from tissue with lung adenocarcinoma showed well demarcated cancer area (Figure 3). Three-dimensional tissue images were obtained using a 3-dimensional image processing software called the Medical Imaging Interaction Toolkit (MITK) developed by German Cancer Research Center (Figure 4).
We compared images from LM examinations and those from synchrotron radiation. Intra-alveolar hemorrhage which were presented as red spots in H & E image, were observed in SR image as gray to whitish spots contained in alveolar spaces which were originally filled with air. Images of adenocarcinoma showed thickened interalveolar septa and consolidation, which were clearly distinguished from hemorrhagic lesions in normal lung tissue The SR images (with spatial resolution of 15μm) were closely similar with x100 magnified LM images, however, could not show distinguishable microstructures as much as x400 magnified LM image (Figure 5).
Figure 4. Three-dimensional reconstruction. (a)3D image of normal human lung tissue. (b) That of lung adenocarcinoma. 3D structures of normal lung cancer and adenocarcinoma could be easily observed through 3D reconstruction of tomographic images. The images were obtained using a 3D image processing software called the Medical Imaging Interaction Toolkit (MITK) developed by German Cancer Research Center. Relating videoclips were appended as supplement data.
Figure 5. Comparison with pathologic findings and SR images. (a) Pathologic images of normal lung tissue and that of SR image. Hemorrhage inside normal alveolar structure (small red circle) was identified in x100 magnified image (left side). Normal alveolar structures including alveolar septa and microvasculature were identified in x400 magnified image (large red circle in middle). The SR image in right side were similar with that of x100 magnified LM image. The hemorrhage lesion was identified however, was not clear compared with that of x400 magnified LM image. (b) Pathologic image of lung adenocarcinoma and that of SR image. Destruction of alveolar structure and interstitial thickening was well noticed both in x100 magnified LM image (left side) and SR image (right side). However, hyperproliferation and thickening of alveolar lining cells (large red circle in middle) could not be observed in SR image.
Comment 5:
- Line 205 to 207 is not necessary to mention in discussion because it is mentioned in section 2.
Answer 5: We are really grateful for your comments. This sentence was described in Section2, Materials and methods, therefore duplication would not be necessary. We thought that we had better remove the sentence that the reviewer had mentioned from section 4. Discussion according to the reviewer’s advice.
Changes 5: We removed the duplicated sentence from section 4. Discussion according to the reviewer’s comment. Changes were highlighted with yellowish color in the text.
The experiment was performed in a synchrotron radiation facility called Photon factory located in Tsukuba, Japan.
Comment 6:
- Line 134-135 doesn’t look like a complete sentence.
Answer 6: We sincerely appreciate your comments. We checked and revised the sentence that the reviewer had pointed out. We changed the order of phrases so that it could compose a complete sentence.
Changes 6: We revised the sentence according to the reviewer’s comment. Changes were marked with yellowish color in the text.
Normal structure including pulmonary vessels, bronchioles, alveolar sacs and interaleolar septa. Intra-alveolar hemorrhage was well identified and distinguished from those findings of lung cancer (Figure 2) in the refraction-contrast images of normal lung tissue.
Comment 7:
- One typo is found, form in line 111
Answer 7: Thank you for your comment and we deeply apologize for our terrible mistake. We checked the error in the sentence that the reviewer had mentioned, and fixed it correctly.
Changes 7: We identified the typo ‘form ‘, and fixed it into ‘from’. Corrections were highlighted with yellowish color in the text.
The actual beam size was measured 1160μm horizontally and 72μm vertically. The charge-coupled device (CCD) cameras were used to collect imaging signals from LAA. It has 14.8μm pixel size therefore the spatial resolution was 15μm.
The authors sincerely appreciate the editor’s and reviewers’ valuable comments. We carefully revised our manuscript according to the reviewers’ precious advice. We believe we did our best to improve the quality of our manuscript, and wish our revision have better achievement. If there were anything to be corrected or appended, please let us know and we promise we will make every effort to revise again.
Reviewer 2 Report
Thank you for the consideration and incorporation of my prior comments into the manuscript. This revised manuscript is stronger and more clearly depicts the objectives and limitations of your study. I have no further suggestions.
Author Response
RESPOND TO THE REVIEWERS
The authors would like to sincerely appreciate the Reviewers for careful review of our manuscript and providing us with their comments and suggestion. Those valuable comments were tremendously helpful for improving the quality of our manuscript. We carefully revised our previous manuscript to adjust it according to the reviewers’ advices. The following responses have been prepared to address all of the editor’s comments in a point –by-point fashion.
Reviewer 2:
Thank you for the consideration and incorporation of my prior comments into the manuscript. This revised manuscript is stronger and more clearly depicts the objectives and limitations of your study. I have no further suggestions.
Answer; We are sincerely grate for the reviewer’s the precious comments. The detailed and precise comments have been a great help for improving the quality of our manuscript. We do appreciate the reviewer for the dedication to our work.
The authors sincerely appreciate the editor’s and reviewers’ valuable comments. We carefully revised our manuscript according to the reviewers’ precious advice. We believe we did our best to improve the quality of our manuscript, and wish our revision have better achievement. If there were anything to be corrected or appended, please let us know and we promise we will make every effort to revise again.